# Task Facet Learning: A Structured Approach to Prompt Optimization

## Abstract

Given a task in the form of a basic description and its training examples, prompt optimization is the problem of synthesizing the given information into a text prompt for a large language model. Humans solve this problem by also considering the different facets that define a task (e.g., counter-examples, explanations, analogies) and including them in the prompt. However, it is unclear whether existing algorithmic approaches, based on iteratively editing a given prompt or automatically selecting a few in-context examples, can cover the multiple facets required to solve a complex task. In this work, we view prompt optimization as that of learning multiple facets of a task from a set of training examples. We exploit structure in the prompt optimization problem and break down a prompt into loosely coupled semantic sections. The proposed algorithm, UniPrompt, (1) clusters the input space and uses clustered batches so that each batch likely corresponds to a different facet of the task, and (2) utilizes a feedback mechanism to propose adding, editing or deleting a section, which in turn is aggregated over a batch to capture generalizable facets. Empirical evaluation on multiple datasets and a real-world task shows that prompts generated using UniPrompt obtain higher accuracy than human-tuned prompts and those from state-of-the-art methods. In particular, our algorithm can generate long, complex prompts that existing methods are unable to generate.

## 1 Introduction

Given a task, choosing an input prompt is a key part of optimizing Large Language Model's (LLM) performance (Kojima et al., 2024; Yang et al., 2023). Minor changes in prompt can lead to performance gains or losses, necessitating prompt engineering (Liu et al., 2023). Typically, manually-developed prompts combine task description with a few in-context examples, along with modifiers like chain-of-thought (Kojima et al., 2024). For greater accuracy, human prompt engineers spend considerable time to identify errors with a current prompt, consider the different facets of a task (e.g., counter-examples, explanations, analogies) that may fix those errors, and include them in the prompt. For instance, for a hate speech classification task, in addition to the definition, it may be helpful to specify the facets that lead to hate speech: the context of conversation, identifying intent, and differentiating hate speech from opinions or closely-related concepts such as vulgarity and profanity.

To avoid the above cumbersome manual process, recent work aims to automate the process of generating natural language prompts that are also interpretable. Since language tokens are discrete, this leads to a challenging discrete optimization problem with a combinatorial space of possible outputs. Techniques for prompt optimization can be divided in two categories: *non-directional*, e.g., random search (Zhou et al., 2022; Zhang et al., 2023) and genetic algorithms (Yang et al., 2023; Guo et al., 2023), where the sampling of new input is "random" and does not explicitly aim to reduce error on a train set; and *directional*, where the sampling of new input depends on some error measure on a representative train sample. Recently, more complex methods have been proposed in the second category including reinforcement learning (Zhang et al., 2022a; Deng et al., 2022), updating prompts using feedback from auxiliary LLMs (Hu et al., 2024; Pryzant et al., 2023), and optimizing the input to a small language model that generates the prompt (Lin et al., 2024b; Chen et al., 2024). While all these techniques focus on editing, adding, or deleting parts of a given prompt, they are developed with the goal of obtaining a concise description of the task. None of these focus on ensuring that multiple facets of a task are added to the prompt.

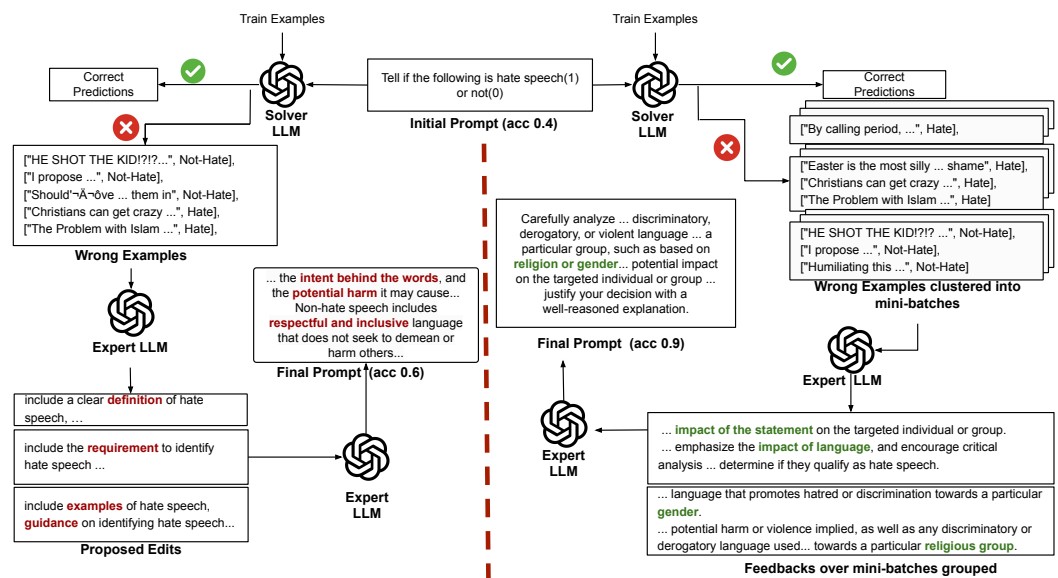

Figure 1: **Existing prompt optimization methods (left) versus UNIPROMPT (right) on the Ethos dataset**: [Left] State-of-the-art prompt optimization methods such as **ProTeGi** (Pryzant et al., 2023) sample from the questions wrongly answered by the current prompt, and use an expert LLM (e.g., GPT-4) to obtain feedback on the mistakes. This approach tends to give very general edits or overfits to specific examples as shown. [Right] In contrast, UNIPROMPT identifies key task *facets* by: (1) clustering examples with similar task facets, and (2) employing a two-tier feedback-based update strategy. The resulting prompt updates extract generalizable concepts from the specific examples.

In this paper, we propose UNIPROMPT, a prompt optimization method to cover diverse, multiple facets of a task and improve overall accuracy. To simulate the manual prompt engineering process, we propose that prompts be constructed from individual *sections*, where each section may correspond to a different facet that humans may consider for the task. Prompt editing proceeds at a section-level: we can add, edit or delete a section from the prompt. Similar to Pryzant et al. (2023); Hu et al. (2024), prompt edits are based on an auxiliary LLM's *feedback* about example predictions with the current prompt. We contribute two key insights in this feedback-based optimization process. First, we find that the feedback on a single example or a randomly selected batch of examples does not yield generalizable facet descriptions. Instead, we propose clustering the inputs beforehand and creating mini-batches such that each mini-batch is sourced from a single cluster. Second, even with clustered batches, the feedback tends to overfit to specific examples or their properties. To generate a prompt edit that conveys a generalizable concept relevant to the task, we propose generating edits at a mini-batch level and then aggregating them at the batch level to yield the final edit (see Figure 1). While the two insights may appear *simple*, we find that they lead to a significant improvement in the extracting diverse facets for a task.

We evaluate UNIPROMPT on several benchmark tasks (QnA, math, code generation) where it consistently achieves higher accuracy than baselines and existing prompt optimization methods. On Ethos, a hate speech dataset, UNIPROMPT obtains 94% accuracy whereas the next best method obtains 82%. Even though UniPrompt focuses only on the instruction and does not include any in-context examples, we find that its instruction-only accuracy is often higher than methods such as DSPy (Khattab et al., 2024) that optimize both. In the few-shot setting, we also compare UNIPROMPT to MedPrompt (Nori et al., 2023), a state-of-the-art prompt composition method. We find that UNIPROMPT, requiring only one LLM call at inference time, obtains the same accuracy as MedPrompt that requires five calls. If we allow multiple calls to UNIPROMPT, we obtain over 4% accuracy gains. Finally, we also evaluate UNIPROMPT on a real-world semantic matching task in a web search engine. Compared to the best manual prompt, the prompt generated from UNIPROMPT leads to over 5% increase in accuracy on the negative class and nearly 2% accuracy increase overall.

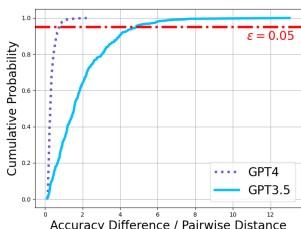 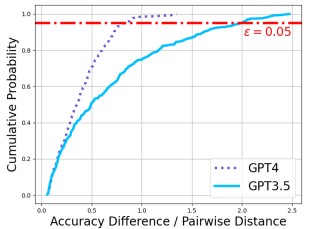 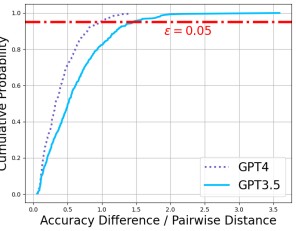

Figure 2: Estimating (probabilistic) Lipschitz constant of models (Definition 1) on (left) Ethos (middle) GSM8K and (right) MedQA datasets for GPT-4 and GPT-3.5 models.

## 2 TEXT-GRADIENT BASED PROMPT OPTIMIZATION: CHALLENGES, INSIGHTS

State-of-the-art prompt optimization methods such as ProTeGi (Pryzant et al., 2023) and TextGrad (Hou et al., 2023) iteratively optimize the prompt for a given task. They adopt the following procedure at a high-level: (1) start with an initial prompt and a training dataset of ⟨question, answer⟩ pairs for the task, (2) randomly sample from the questions wrongly answered by the current prompt to form a batch, (3) use an expert LLM to obtain feedback on the random batch, (4) apply the feedback to the prompt. This procedure is illustrated in Figure 1 [Left]. In this section, we study when such optimization is feasible and what may be the issues with current prompt optimization procedures.

### 2.1 WHEN IS DIRECTIONAL TEXT OPTIMIZATION FEASIBLE?

Consider the class of sequential algorithms as outlined above. The objective is to improve the accuracy of a given (black-box) *solver LLM* $f : \mathcal{X} \to \mathbb{R}$ that takes as input a prompt $\mathbf{x} \in \mathcal{X}$ and outputs the average accuracy on a validation set $D_v$. Since the set of prompts is combinatorially large, we assume that all prompts can be embedded in a vector space such that distance between two prompts in the space correspond to their semantic similarity. The prompt optimization problem can be written as $\arg\max_{\mathbf{x} \in \mathcal{X}} f(\mathbf{x}; D_v)$.

Previous work has shown that LLMs can be brittle to their input: changing the prompt slightly can create a significant difference in performance (Zhuo et al., 2023). We want to understand if the optimization problem is well-conditioned. Typically, conditioning can be determined by the Hessian. However, since $f$ is black-box, we approximate it by measuring sensitivity, or more specifically, Lipschitz continuity near the optimal solution. Based on prior work on defining continuity of neural networks (Mangal et al., 2020), we use a probabilistic notion.

**Definition 1** *Probabilistic Lipschitz Continuity (Mangal et al., 2020). Given a probability distribution over inputs $\mathcal{X}$, $r \geq 0$, and a distance measure $d$ such as $\ell_1$ or $\ell_2$ norm, a function $f : \mathcal{X} \to \mathbb{R}$ is $(L, \epsilon)$-probabilistically Lipschitz with constant $L \geq 0$, if*

$$\Pr_{\mathbf{x}, \mathbf{x}' \sim \mathcal{X}}[\mathrm{d}(f(\mathbf{x}), f(\mathbf{x}')) \leq L. \, \mathrm{d}(\mathbf{x}, \mathbf{x}') \mid \mathrm{d}(\mathbf{x}, \mathbf{x}') \leq r] \geq 1 - \epsilon. \quad (1)$$

Note the focus on small changes in input through the parameter $r$. Intuitively, the Lipschitz property bounds the maximum change in $f$ given a small change in input prompt. Typically, the lower bound of error for any sequential optimization algorithm over $f$ is directly proportional to the Lipschitz constant $L$ (Malherbe & Vayatis, 2017). Therefore, for faster convergence, it is desirable to have a low $L$, especially near the optimal solution.

Empirically, we estimate $L$ by sampling task-relevant prompts so that they are close to the optimal solution. Then we make small changes to the prompt such that the semantic meaning stays the same and measure the change in $f$ (See Appendix A.1 for experimental details). We show the change in $f$ per change in input for GPT4 and GPT3.5 models in Figure 2 for the Ethos, GSM8K and MedQA datasets. Assuming $\epsilon = 0.05$, probabilistic Lipschitz constant $L$ for GPT4 is $< 1$, whereas it is higher for GPT3.5. Thus, as the model sizes increases, the probabilistic Lipschitz constant decreases. So:

**Observation 1: Larger models are more amenable to prompt optimization.**

## 2.2 CAN WE DO BETTER THAN RANDOM BATCHING?

Batching is standard in gradient-based optimization to obtain robust gradients. Text-gradient based prompt optimization techniques such as ProTeGi (Pryzant et al., 2023) and TextGrad (Hou et al., 2023) also adopt batching by randomly sampling from training examples where the solver LLM made a mistake. In the early iterations of optimization, there can be many such examples — is random batching sufficient to get meaningful feedback (i.e., the text gradient) from the expert model?

To investigate this, we consider the Ethos dataset where the task is to classify text as hate speech or not. We start with a basic task description as the prompt, "*Is the following text hate speech (1) or not (0)?*", run it with GPT-3.5 on 200 examples, and then provide random batches of incorrect predictions to GPT-4 to identify prompt edits. GPT-4 is prompted to provide edits to the current prompt such that the errors are minimized. With a random batch, the feedback obtained is relevant for the task, but fails to identify specific concepts. Example feedback include,

> *The instruction should include a clear definition of hate speech...*
> *The instruction should include examples of hate speech, guidance on identifying hate speech...*

Instead, if we cluster the incorrect examples (see Section 3.1) and create clustered batches, we obtain the following feedback.

> *The instruction should include..potential harm or violence implied, as well as any discriminatory or derogatory language used...towards a particular religious group.*
> *The instruction should include ...think about the impact of the statement on the targeted individual or group.*
> *The instruction should include...language that prompts hatred or discrimination towards a particular gender.*

The first feedback captures the religious and harm-based aspect of hate speech whereas the second captures the aspect of measuring impact on the targeted entity. This case-study suggests that the same LLM is able to identify different facets due to clustered batches.

**Observation 2: Clustered-batching improves the quality of text gradients.**

## 2.3 CAN WE LEARN A GENERALIZABLE PROMPT FROM A LIMITED SET OF EXAMPLES?

The optimal prompt should consider all the aspects of the task. So, any prompt optimization algorithm (or a human prompt engineer for that matter) may have to exhaustively sample from the data distribution in many real-world scenarios. In practice, however, we have access only to a limited number (a few hundreds) of training examples (as it requires exhaustive manual labeling). We present such a case study in Section 4.4 that arises in search and recommendation pipelines. The task is to infer if two search queries share identical intent or not. Here, the notion of "identical intent" is captured implicitly in the human-labeled query-pairs.

In this data-constrained setting, consider the standard in-context learning (Brown et al., 2020), where the prompt is composed of a simple task description and a set of labeled examples from the task. There are many ways in the literature to select examples for in-context learning, including random sampling and k-nearest neighbors. In our setting, perhaps it would be beneficial to consider the marginal utility of examples (Zhang et al., 2022b; Gupta et al., 2023a), i.e., add examples where the model fails rather than ones where the model already succeeds (see Table 8 in Appendix). This suggests that one could use a greedy algorithm for iteratively optimizing the prompt by finding failing examples and adding to the prompt. *But would such a procedure converge to a good solution that generalizes to unseen examples?*

As we saw in the example above, such a procedure based on incorrect examples does provide specific facets, such as being focused on religious or gender groups. However, adding any of these texts directly to the prompt may be too specific and may miss out on other groups that may be targeted by hate speech (e.g., groups based on ethnicity). A procedure to aggregate the feedback may encourage

the expert LLM to propose edits that are generalizable. In this example, we provide the three feedback texts above to the same expert LLM and ask it to summarize them into a single feedback. We obtain,

> *The instruction should...consider whether the statement contains discriminatory, derogatory, or violent language that promotes hatred or harm towards a particular group, such as based on religion and gender.*

As can be seen above, the summarized feedback captures the essence of the individual feedback texts and generalizes it to any group. A two-tier feedback, therefore, can help in distilling important aspects of the task implicit in the examples, rather than directly use or rephrase the (limited) examples.

**Observation 3: Two-tier feedback helps learn generalizable facets in prompts.**

## 3    UNIPROMPT: GENERATE A PROMPT TO CAPTURE MULTIPLE TASK FACETS

Our observations above indicate that collecting feedback over individual examples or randomly sampled batches may lead to memorization of individual examples (see Figure 1) rather than recognition of facets that are important to the task. To overcome these shortcomings, the proposed method, UNIPROMPT, makes two contributions. First, we follow a two-tier setup of synthesizing feedback for a batch of training examples. We break up a batch into mini-batches, collect feedback on each of the mini-batches and then use a separate prompt to aggregate the different feedback texts into a generalizable concept. Second, to increase chances that a mini-batch corresponds to a coherent facet, we cluster the training data beforehand and ensure that each mini-batch consists of examples from the same cluster. We provide details of the algorithm below (see Algorithm 1).

The algorithm receives as input a one-line task description and a set of ⟨question, answer⟩ demonstrations. It extracts key concepts or facets relevant to the task and updates prompt sections using them, with the overall goal of increasing validation accuracy. Our analysis in Section 2 indicates that such a directional optimization procedure based on iteratively adding task facets can converge to a good solution, especially for solver LLMs such as GPT-3.5 and GPT-4.

**Notation:** We denote the training set of $N$ examples with $D_t$ where each example is a question-answer pair of the form $(q_i, a_i)$; and the validation set of $K$ examples with $D_v$. Input to the algorithm is the solver LLM $f_{LLM}$, train set $D_t$, validation set $D_v$, initial prompt for the task $p_0$, one-line task description $T$. In addition, we assume access to an "expert LLM" such as GPT-4.

### 3.1    TASK FACET LEARNING USING EXAMPLES

Extracting task-relevant concepts from a set of examples to refine a prompt is a complex problem comprising multiple steps. Given a set of incorrect predictions, one needs to analyze what went wrong in each prediction, form hypotheses, aggregate the hypotheses to identify specific concepts that are relevant for the task. Then, for each concept, one needs to attribute which facet/section of the current prompt needs to be edited to incorporate the concept. These operations are highly model-specific and are difficult to execute reliably. Therefore, we exclusively rely on an expert LLM.

First, we prompt the expert LLM to diagnose mistakes (*feedback*) in each example given the answer and chain-of-thought reasoning produced by the solver LLM. Subsequently, we use this feedback to generate precise edits for the prompt that may fix the error. These individual edits are then aggregated over a mini-batch and fed back into the same LLM, which then identifies a few major edits to be applied to the current prompt. To aid in identifying major edits that correspond to generalizable facets, we propose to cluster the examples as a preprocessing step and create clustered batches, such that each cluster shares some common facet of the task.

#### 3.1.1    PREPROCESSING STEP: CLUSTERING EXAMPLES TO FACILITATE FACET IDENTIFICATION

We explore two approaches for clustering: *topic-based clustering*, and *feedback-based clustering*.

**Topic-Based Clustering.** Given a set of examples, we identify $l$ topics spanning the entire train set. This type of clustering is motivated by the observation that solver LLM may make similar mistakes on examples from the same topic. Hence, for such examples, a common edit to the prompt could improve predictions for all the examples. To obtain the clusters, the expert LLM is prompted (for

---

**Algorithm 1:** UNIPROMPT

---

**Input:** Train set $D_t$, validation set $D_v$, initial prompt for the task $p_0$, one-line task description $T$
**Output:** Optimized prompt $P^*$ for the given task

1 Cluster train set, initialize history and validation accuracy arrays: $C \leftarrow \texttt{cluster}(D_t)$, $H \leftarrow \{\}$, $V \leftarrow []$;
2 Initialize a beam of size 2 with the initial prompt: $p_1 \leftarrow p_0$ and $p_2 \leftarrow p_0$
3 **for** *each cluster c in C* **do**
4     $B \leftarrow \texttt{batches}(C)$;
5     **for** *each batch $b \in B$* **do**
6         $M \leftarrow \texttt{mini-batches}(B)$
7         $F \leftarrow []$
8         **for** *each mini-batch $m \in M$* **do**
9             Evaluate the best prompt on mini-batch: $a_m \leftarrow \texttt{LLM}(m, p_1)$
10             Get feedback from the expert given history of mini-batch, accuracy and task description: $f \leftarrow \texttt{Feedback}(T, a_m, H[m])$
11             $F.\texttt{insert}(f)$
12         Combine feedbacks over a batch: $F_b \leftarrow \texttt{Combine}(F)$
13         Apply feedback to get updated prompts; $q_1 \leftarrow \texttt{apply}(F_b, p_1)$; $q_2 \leftarrow \texttt{apply}(F_b, p_2)$
14         Update the beam: **if** *not($p_1 = p_0$)* **then**
15             $p_2 \leftarrow \texttt{second-high-acc}([p_1, p_2, q_1, q_2], b)$
16         $p_1 \leftarrow \texttt{highest-acc}([p_1, q_1, q_2], b)$
17     Evaluate the best prompt on validation set: $acc_v \leftarrow \texttt{evaluate}(p_1, \ D_v)$
18     $V \leftarrow V.\texttt{append}(acc_v)$
19     Stop based on the validation accuracy: $c \leftarrow \texttt{early-stop-criteria}(V)$
20     **if** $c$ **then**
21         break

22 **return** $p_1$;

---

prompt see Appendix A.6) to provide a broad sub-topic $t_i$ for each question. Then the resultant list of sub-topics $\{t_1, t_2, \ldots, t_N\}$ is again clustered into $k$ topics $\{t'_1, t'_2, \ldots, t'_l\}$ by prompting the expert LLM. Based on this clustering, each example $q_i, a_i$ is assigned a cluster $t'_j$ corresponding to $t_i$.

**Feedback-Based Clustering.** Another way to find examples that share similar task facets is the feedback they receive based on the initial prompt's predictions. For instance, for a physics-based task, if two examples from different topics obtain the same feedback from the expert LLM to edit the "*Rules*" section of the prompt and include the statement, "*Draw all forces on each body before writing the equations*", then we argue that such examples can be clustered. This type of clustering makes the broad edit identification step easier. To obtain the clusters, we first evaluate all training examples against the initial prompt $p_0$ and store the feedback $f_i$ from the expert LLM, corresponding to each incorrectly answered example $q_i, a_i$ (all the correctly answered questions form one cluster). We then prompt the expert LLM to cluster these feedbacks $\{f_1, f_2, \ldots, f_N\}$ into $l$ clusters (see Appendix A.7). For each cluster, we create a batch of examples $q_i, a_i$ corresponding to feedbacks in that cluster.

### 3.1.2 GENERATING PROMPT EDITS THAT GENERALIZE TO MULTIPLE EXAMPLES

**Two-tier Feedback.** To encourage generalizable feedback from the expert LLM, we obtain feedback at two levels: mini-batch and batch. Given a batch (created using clustering discussed above), we break it up into mini-batches. For each mini-batch $m$, we construct a prompt consisting of incorrectly-answered questions in $m$, the chain-of-thought produced by the solver LLM, their incorrect predictions and the ground-truth answers. We ask the expert to provide one feedback for the mini batch (prompt is provided in Appendix A.8). The expert can suggest the following edits: add a section or subsection, delete a section or subsection, and edit a section or subsection.

Given the different edits for mini-batches within a batch $b$, we invoke the expert LLM again to summarize these edits into a single section update. This combination ensures some degree of smoothness at every update which helps stabilize training. To make sure that the expert is able to

generate generalizable edits, we additionally provide a random set of incorrect examples that are not in the current batch and ask it to suggest an edit based on the existing edits that can correct the errors. As before, the class of edits allowed is the same.

**History for effective exploration.** To ensure comprehensive, non-repetitive exploration of prompts, we also provide the batch-level history of edits (Hu et al., 2024; Yang et al., 2023) in the mini-batch-level prompt. History $H[b]$ is presented as $\{e_i, acc_i - acc_{i-1}\}$ where $e_i$ is the edit proposed at the $i^{th}$ update and $acc_i$ is the accuracy of the $i^{th}$ updated prompt (See Appendix A.8 for the full prompt).

### 3.1.3 EDITING THE PROMPT

Once the final set of edits is received for a batch, we use the expert LLM to apply edits to the current prompt (see Appendix A.9 for the prompt). An edit is accepted only if it increases the validation accuracy compared to the current prompt. We call this approach *"Greedy"*. Alternatively, we maintain a beam of 2 best performing prompts based on validation accuracy, apply the edit to the two prompts, and update the beam to retain the top 2 performing prompts. We call this method *"Beam"*. To avoid overfitting on the train examples (or keep adding unnecessary information to the prompt), we employ early stopping in the optimization process (more details in Section 4).

### 3.2 PROMPT INITIALIZATION

Our first option is to initialize the prompt using only the task description, i.e., $p_0$ has a single section titled *Introduction* containing the input task description. Second, we finetune Llama2-13B model to generate a prompt with sections such as Introduction, Tricks, and Corner Cases, similar to the initial prompt that a human prompt engineer may produce. To finetune, we use GPT-4 generated data consisting of (task description, section title, section contents) triples. Details are in Appendix A.3.

Examples of these two kinds of prompts for different tasks are in Appendix A.2.

### 3.3 EFFICIENCY CONSIDERATIONS: COMPUTATIONAL COMPLEXITY

We now consider the compute complexity of the UNIPROMPT algorithm for a given task in terms of the number of expert or solver LLM calls made per epoch, stage-wise.

**Clustering**: First, we evaluate all the training examples using the current prompt. Second, for every wrongly predicted example, we obtain feedback from the expert LLM. Third, for the given set of feedbacks, we use a single call to cluster it into $l$ clusters. Each of the above steps incurs $O(N)$ queries, so the total query complexity of the clustering stage is $O(N)$. Finally, for each example, i.e., (question, answer) pair, we simply map it to the $l$ clusters (no LLM calls). This is a one-time cost.

**Mini-batch feedback and Batch-level aggregation**: At a given epoch, we evaluate every question in the mini-batch using the current prompt and the solver LLM ($N$ queries overall). Next, we obtain one feedback over all the wrong questions in the mini-batch $m$ ($N/|m|$ queries). We use one call to aggregate these feedbacks. For prompt selection, we evaluate 4 prompts on the batch $b$ (2 per beam), so $O(4|b|)$ queries per batch. Hence overall query complexity is $N + N/|m| + 4N + 1$ or $O(N)$.

Assuming LLM throughput of 0.5 queries per second, a training + validation set of 300 examples, 10 clusters, and 20 epochs, it takes under 7 hours to train.

## 4 EXPERIMENTS

**Datasets** We perform comprehensive evaluation on four standard datasets : (1) Ethos (Mollas et al., 2020), (2) ARC (Clark et al., 2018), (3) MedQA (Jin et al., 2021), and (4) GSM8K (Cobbe et al., 2021). Ethos, ARC, and MedQA contain multiple choice questions, and GSM8K contains questions with integer answers. In addition, we also evaluate UNIPROMPT on the medical QnA datasets used in the MedPrompt (Nori et al., 2023) work; as well as two popular code generation datasets, HumanEval (Chen et al., 2021) and MBPP (Austin et al., 2021).

**Implementation details** We set the initial prompt $p_0$ for each task as the one-line task description and use 200 examples as the train set and 100 examples as the validation set for all the compared methods. We use GPT-3.5-Turbo as the solver model. For `Feedback` and `Combine` in UNIPROMPT, we use

Table 1: Test accuracies (%) of the compared methods with GPT-3.5-Turbo as the solver model in the zero-shot setting (**best** in bold; second best underlined;). The two UNIPROMPT rows correspond to our proposed method. We also include few-shot methods in the last two rows (DSPy variants) for comparison; *\***best** in bold to distinguish the few-shot setting.

| Method | Ethos | ARC | MedQA | GSM8K |
|---|---|---|---|---|
| Task Description | 76.8 | 79.7 | 52.7 | 59.4 |
| Expert Prompt | 74.1 | 78.4 | 53.1 | 78.9 |
| Llama Prompt (Section 3.2) | 74.0 | 89.7 | 52.6 | 79.5 |
| CoT | 72.0 | 79.4 | 50.3 | 76.3 |
| OPRO | 65.4 | 79.1 | 53.3 | 77.1 |
| ProTeGi | 76.0 | 78.8 | 52.9 | 77.3 |
| Evoke | 63.5 | 89.0 | 52.8 | 81.0 |
| EvoPrompt | 81.6 | 89.9 | 50.3 | 81.4 |
| DSPy (`MIPRO v2`, zero-shot) | 79.7 | 82.8 | **61.9** | 77.3 |
| TextGrad | 79.5 | 76.5 | 50.6 | 81.6 |
| UNIPROMPT (Init = Task Description) + Beam | 92.3 | 86.0 | 57.1 | **82.4** |
| UNIPROMPT (Init = Task Description) + Greedy | **93.7** | **90.5** | 55.5 | 82.3 |
| DSPy (`BootstrapFewShotWithRandomSearch`) | 86.6 | 87.5 | *\*68.5 | 74.3 |
| DSPy (`MIPRO v2`, few-shot) | 84.0 | 86.0 | 62.9 | 79.7 |

GPT-4 as the expert (see ablation in Section 4.5). We maintain a beam size of 2. Mini-batch sizes (and batch sizes) are constrained by the context length of GPT-4. We find that mini-batch sizes 3 to 5 and batch sizes 5 to 7 work the best for our datasets. The temperature of the LLMs for our method is set to 0 for reproducibility of results. We employ early stopping at batch-level in UNIPROMPT.

**Baselines** We compare UNIPROMPT with the following techniques and baselines: (1) **Task Description**: prompt is the one line task description that we use to initialize UNIPROMPT; (2) **Chain-Of-Thought** (or CoT) prompting (Kojima et al., 2024); (3) **Expert Prompt**: the prompt optimized by humans taken from prior works (Nori et al., 2023); (4) **OPRO** (Yang et al., 2023), that uses LLMs for discrete optimization over text prompts; (5) **ProTeGi** (Pryzant et al., 2023) that proposes textual gradients and selects edits to prompts using bandit techniques; (6) **Evoke** (Hu et al., 2024) that uses two instances of LLM, one that scores the current prompt, and the other that edits the prompt; (7) **EvoPrompt** (Guo et al., 2023) that uses genetic algorithms to search through the space of prompts; (8) **TextGrad** (Hou et al., 2023), state-of-the-art framework for automatic differentiation of prompts via text; (9) **DSPy** (Khattab et al., 2024), a recent programming model for optimizing LLM prompts; and (10) **MedPrompt** (Nori et al., 2023), a state-of-the-art prompt composition method.

### 4.1 PERFORMANCE OF UNIPROMPT COMPARED TO EXISTING METHODS

We start with the zero-shot setting, where we do not include labeled examples in the prompt for any of the compared methods. We report results for two versions of our method in Table 1, which differ in the combining strategy (from Section 3.1.3)—beam search vs greedy.

UNIPROMPT variants significantly outperform the baselines including CoT and the state-of-the-art prompt optimization techniques like ProTeGi that crucially leverage LLMs for performing iterative prompt edits. On three out of four datasets, UNIPROMPT achieves the best test accuracies among all the methods. We achieve maximum gains on the Ethos dataset with a $18.2\%$ increase in accuracy over the expert prompt. Further, we see accuracy increases of $4.0\%$ on MedQA, $3.5\%$ on GSM8k, and $7.6\%$ on ARC-Challenge datasets. We show UNIPROMPT training behavior in Appendix A.16.

We also present comparisons to state-of-the-art DSPy method in the few-shot setting (8 `bootstrapped_demos`) using two optimization settings provided by their framework. The last two rows of Table 1 show that UNIPROMPT in the zero-shot setting convincingly outperforms DSPy in the few-shot setting, on three out of four datasets.

## 4.2 COMPARISON WITH MEDPROMPT

MedPrompt (Nori et al., 2023) is a recent, competitive prompting technique without any training component. It employs three key ingredients: (1) few-shot prompting, where five relevant examples are selected using k-nearest neighbors (kNN); (2) CoT reasoning on the selected examples; and (3) self-consistency and ensembling with option shuffling at inference time. They evaluate on 4 medical datasets (that none of the competing methods in Table 1 evaluate on) using GPT-4 as the solver model. So, we compare UNIPROMPT in the same setting in Table 4. UNIPROMPT (first row), which requires only one call at inference time, performs almost as well as MedPrompt (last row), which requires five calls, on three out of four datasets. As we incrementally add kNN few-shot, CoT, and ensembling to our prompt, we see a significant increase in accuracy of 4.35% on average across all datasets.

## 4.3 PERFORMANCE ON GENERATION TASKS

Our evaluations so far have been on multiple-choice QnA, math, and classification datasets. We now evaluate UNIPROMPT on generation tasks; specifically, generating code given a natural language specification. We use HumanEval (Chen et al., 2021) and MBPP (Austin et al., 2021) datasets consisting of Python coding problems. We initialize with a simple prompt, "*You are a software engineer. You are given a function signature and a description of the function. You have to complete the function.*" We use GPT-4-Turbo as both the solver and the expert LLM.

HumanEval does not have train or validation sets. So, we take random 100 examples from MBPP to train a prompt for HumanEval. Similarly, for MBPP, we take random 50 examples from HumanEval as train set. We evaluate the final prompts on HumanEval and MBPP test sets. The results are given in Table 2. The metric is % of solved coding problems (evaluated using the provided test cases) in the datasets. The prompts produced by UNIPROMPT outperform standard prompting of LLMs.

Table 2: Performance (% solved problems) of UNIPROMPT (GPT-4-Turbo solver) on code generation datasets, compared to GPT-4 (OpenAI et al., 2023) and newer models.

Table 3: Ablation on solver and expert LLM choices for UNIPROMPT on the Ethos dataset. 'Init' and 'Final' denote initial (i.e., task description) and final prompt accuracies.

| Method | HumanEval | MBPP |
|---|---|---|
| GPT-4 | 67.0 | 87.5 |
| GPT-4-Turbo | 87.1 | 90.9 |
| GPT-4o | 90.2 | 92.4 |
| UNIPROMPT | **93.8** | **92.5** |

| Expert LLM | Solver LLM | Init | Final |
|---|---|---|---|
| GPT-3.5-Turbo | GPT-3.5-Turbo | 76.8 | 82.4 |
| GPT-4 | GPT-3.5-Turbo | 76.8 | 92.3 |
| GPT-3.5-Turbo | GPT-4 | 89.8 | 91.4 |
| GPT-4 | GPT-4 | 89.8 | **94.3** |

## 4.4 RESULTS ON A REAL-WORLD TASK: SEARCH QUERY INTENT

The task of inferring if two search queries share identical intent or not arises in search and recommendation pipelines, and is tackled today using LLMs. It is challenging because it requires domain knowledge (e.g., brands and product categories), and depends on cultural and geographical biases (e.g., "cricket" and "cricket game" are likely to mean the same in UK, but unlikely in the US). So, examples are crucial for understanding the task and engineering a prompt that generalizes well.

We sample real user queries from a proprietary application, rewrite them using ML models, and ask expert judges to label the query-pairs as identical or otherwise based on prescribed guidelines. We use a set of 200 examples as training data, and an additional 50 examples as validation set, to learn a prompt using UNIPROMPT, starting from the one-line description: *Tell if query A and query B have same intent or not*. The dataset is heavily biased towards positive samples, so the metric of success is improvement in accuracy, over the best manually-engineered prompt, on the positive and negative classes individually. For testing, we use a separate labelled set of 2527 examples from two geographies — one where the training data was sampled from, and the other unseen.

The prompt obtained using UNIPROMPT improves over the best manual prompt by 5.77% on the negative (rare) class, by 0.23% on the positive class, and by 1.86% overall on the test set. The learnt prompt captures the following facets of the task: (1) recognizing variations in names and abbreviations, and how they do not necessarily change the context; (2) recognizing the specificity of

brands, and how even minor variations do change the context; and (3) recognizing the specificity of terms in queries, and how lack of specific terms can indicate departure of intent.

### 4.5 ABLATIONS

**Impact of Clustering, Inclusion of History, and Greedy Update**    The results are shown in Table 6. We see that clustering as well as edit history components (Section 3.1) are critical for performance of UNIPROMPT in all the datasets. We see a major drop of $14.8\%$ in accuracy in the Ethos dataset when clustering is removed, and a $4.3\%$ drop when history component is removed. In all the datasets except GSM8K, we find clustering is more important than history. This can attributed to limited variability of question types (all grade-8 arithmetic) in GSM8K than in others.

We also find that the greedy update rule (Section 3.1.3) proves to be superior or competitive compared to beam search in relatively easier datasets — where even less exploration produces good results, greedy proves to be a more effective update rule. On the other hand, in more complex datasets like MedQA, greedy appears to be a bad strategy. We also see that clustering examples based on feedback ("Fb Clustering") is a better strategy than clustering based on topics, except for the Ethos dataset.

**Impact of initial prompt**    We consider three initializations of UNIPROMPT in Table 5. A simple one-line task description initialization for UNIPROMPT achieves the best accuracy on three out of four datasets. On the ARC dataset, initializing with the prompt generated by the Llama2-13B model (as discussed in Section 3.2) gives significant improvement over other initializations.

**Expert model capacity**    UNIPROMPT is not restricted to improving the prompts for less capable (solver) LLMs using more capable LLMs as experts. In Table 4, we showed how UNIPROMPT works very well with GPT-4 as both the solver and the expert LLM. Table 3 summarizes the results for different choices of expert and solver LLMs, on the Ethos dataset: UNIPROMPT improves the prompt for GPT-4 with less-capable GPT-3.5-Turbo as the expert, achieving a competitive $91.4\%$ accuracy.

## 5    RELATED WORK

Here, we highlight relevant work that are not addressed in the manuscript so far. Deng et al. (2022) present a discrete prompt optimization method, RLPrompt, using reinforcement learning, where a policy network learns to generate effective prompts through reward-based training, with an emphasis on enhancing training efficiency through effective reward stabilization techniques. A drawback of such automatic prompt optimization approaches (Pryzant et al., 2023; Zhou et al., 2022; Deng et al., 2022; Yang et al., 2023) is that the prompts generated tend to be short, often comprising only one or two sentences, which may not fully encapsulate the complexity of the task at hand.

Another recent line of work leverages human feedback in prompt optimization. Automated Prompt Optimization with Human Feedback  (Lin et al., 2024a) optimizes prompts for black-box LLMs using human preference feedback. Besides the obvious overhead, it might also introduce potential biases.

Prior research (Wei et al., 2023; 2024) has highlighted the significance of specific sections within prompts. However, existing methods do not specifically target the optimization of individual sections and their respective contents within the prompts. Hsieh et al. (2023) investigate the use of greedy and genetic algorithms to edit lengthy prompts. Their method focuses on paraphrasing one line at a time starting from an existing prompt, compared to our goal of learning facets of a task from scratch, thereby compromising generalization accuracy (see Appendix A.12). Another orthogonal line of work explores algorithmic selection of in-context examples (Min et al., 2022; Gupta et al., 2023b; Wu et al., 2023; Srivastava et al., 2024; Sun et al., 2024).

## 6    CONCLUSIONS

We presented a method inspired by the human prompt engineering process to generate complex prompts from scratch that include different facets of a task. Our algorithm provides significant improvements over baseline prompt generation methods on multiple standard datasets. Just like in-context learning (Ji et al., 2024), task facet learning could also benefit from connections to submodular optimization (Krause & Golovin, 2014). We leave this as future work.

## 7 REPRODUCIBILITY STATEMENT

Our source code will be made available as open-source upon receiving the necessary approvals. For the other baselines, we have used the code that was open-sourced by the respective authors.

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

# A APPENDIX

## A.1 DETAILS ON ESTIMATION OF LIPSCHITZ CONSTANT $L$

TO calculate the Lipschitz constant for a given LLM and task, we take a human written promp and generate it's paraphrases using GPT-4. We prompt GPT-4 with the following text: "You are given a sentence, you have to generate 30 paraphrases of the sentence, make sure that the cor content of each paraphrase is same, you can use add, subtract or change words". These paraphrases are then

Table 4: Comparison of UNIPROMPT ("Ours") with MedPrompt, with GPT-4 as the solver model.

| | MedQA | PubMedQA | MedMCQA | MMLU MG |
|---|---|---|---|---|
| Ours | 80.9 | 70.3 | 79.2 | 78.0 |
| Ours + kNN | 81.0 | 72.2 | 81.4 | 94.0 |
| Ours + kNN + CoT | 83.9 | 74.7 | 82.6 | 96.0 |
| Ours + kNN+ CoT + Ensemble | **87.0** | **75.6** | **84.5** | **99.0** |
| MedPrompt | 80.6 | 71.2 | 79.1 | 98.0 |

Table 5: Ablation on the initial prompt for UNIPROMPT (**best** test accuracy in bold).

| Init Prompt | Ethos | ARC | MedQA | GSM8K |
|---|---|---|---|---|
| Expert Prompt | 84.0 | 86.0 | 52.3 | **82.4** |
| Llama Prompt | 92.0 | **90.5** | 55.5 | 81.5 |
| Task Description | **92.3** | 86.0 | **57.1** | **82.4** |

evaluated on the validation set $D_v$. For a measure of distance between two prompts, we take the cosine similarity between the embeddings of two prompts. We use text-ada-002 for generating the text embeddings for prompts.

## A.2  PROMPT INITIALIZATION

**One line task descriptions:**

1. Ethos: In this task, you have to determine whether a given text is hate speech or not.

2. ARC: You have to solve the following science question.

3. GSM8K: In this task, you are given a math question. You have to solve the question.

4. MedQA: In this task, you are given a medical question. You have to solve the question.

**An example of sectioned initialization prompt generated using finetuned Llama Model**

```
Introduction:
Assume the role of a science expert and answer the given
question by selecting one of the options A, B, C or D.
        1. Understand and solve science questions by selecting
        the best answer from a given list of options.
        2. Identify the logic behind the choices provided and
        make an informed decision.
        3. Use contextual clues to choose the most accurate
        answer.
        4. Be aware of the differences between science and
        everyday language.

Task Description:
        Scientific inquiry: Science is the systematic study of
        the structure and behavior of the physical and natural
        world through observation and experiment. The
```

Table 6: Ablation of design choices in UNIPROMPT with GPT-3.5-Turbo as the solver model.

| | Ethos | ARC | MedQA | GSM8K |
|---|---|---|---|---|
| UNIPROMPT − History | 88.0 | 84.6 | 55.3 | 80.8 |
| UNIPROMPT − Clustering | 77.5 | 82.0 | 54.1 | 81.5 |
| UNIPROMPT | 92.3 | 86.0 | 57.1 | 82.4 |
| UNIPROMPT + Greedy | **93.7** | 90.5 | 55.5 | 82.3 |
| UNIPROMPT + Fb Clustering | 87.2 | **91.2** | **58.3** | **82.5** |

scientific method is a process for acquiring knowledge that has been improved upon since its inception in the 17th century. It involves making observations, formulating hypotheses as to their causes, and experimenting with them to support or refute the hypotheses.

Real-life Application:
1. Assisting Students in Science Classes:
    In the context of science education, the ability to solve science questions can help students to better understand and internalize the concepts. By familiarizing themselves with the basic principles of science, students can develop a stronger foundation of knowledge.
2. Improving Scientific Literacy:
    Scientific literacy is a critical skill in today's world, where scientific knowledge is increasingly important. By solving science questions, individuals can improve their understanding of scientific concepts and be more informed about scientific developments.
3. Scientific Questions:
    In daily life, there are many questions that require scientific knowledge to answer. For example, understanding the science behind certain phenomena, such as why a magnet sticks to a refrigerator door, can help us in our day-to-day life.
4. Increased Awareness:
    By answering scientific questions, we can develop a deeper understanding of the world around us and increase our awareness of scientific phenomena. This can help us in our daily lives and make us more knowledgeable individuals.

Background Knowledge:
    1. Understanding of the basic concepts of science and physics, such as the difference between heat, temperature and friction.
    2. Basic knowledge of the different types of skin surfaces, such as dry, wet, rough, smooth, etc.
    3. Familiarity with the different types of magnets and their properties.
    4. Understanding of the different factors that affect the adhesion of magnets to different surfaces.
    5. Knowledge of the different types of sedimentary rocks and their properties.

Challenges:
1. Ambiguity in the question:
    The question might be ambiguous in nature, and it can be difficult to understand the exact meaning of the question. In such cases, it is important to read the question carefully and identify the key concepts or keywords. This can help in arriving at the correct answer.
2. Scientific terms or concepts:
    The question might contain scientific terms or concepts that are unfamiliar to the user. In such cases, it is important to understand the meaning of these terms or concepts and their relationship with the question.
3. Difficulty in understanding the question:

```
        Sometimes, the question might be complex or abstract, making
        it difficult to understand or interpret.
4. Misleading statements or information:
        The question might contain misleading or false information,
        making it difficult to determine the correct answer.
5. Contradiction:
        The answer can be in conflict with well-known scientific
        facts or principles. In such cases, it is important to make
        a careful analysis of the evidence and choose the answer
        that is most consistent with the available

Simplification:
1. Identify the key elements in the question:
        Ask yourself, "What is the main question in the question?"
        Identify the key elements and focus on them to solve the
        problem.
2. Understand the context:
        Understand the context of the question and the background
        knowledge you need to answer it.
3. Identify the answer choice:
        Identify the answer choice that best fits the context and
        background knowledge.
4. Eliminate the distractors:
        Eliminate the distractors that don't fit

Tricks:
1. Read the question carefully: Understand the question and its
context. This will help in understanding the information and
concepts needed to solve the question.
2. Identify the key concepts: Identify the key concepts and
keywords in the question. This will help in understanding the
main idea and focus on the relevant information.
3. Understand the question structure: Understand the structure
of the question. This will help in identifying the appropriate
answer option and avoiding distractions.
4. Look for clues: Look for clues in the question and the
answer options
```

### A.3 SLM Training Details

To induce the ability of structured prompt generation in a smaller language model, we curate a section-wise dataset of around 12,000 task-prompt pairs. The tasks for training dataset creation were taken from tasksource library (Sileo, 2023) that contains around five hundred classification tasks. We extract the task description from tasksource-instruct, which contains tasksource dataset recast with instructions. For instance, the task description for BIG-bench Entailed Polarity task is, *"Given a fact, answer the following question with a yes or a no"*. The dataset provides diverse tasks and their short description, but not the human-generated prompts for each task. To approximate human-generated prompts, we use GPT-4 as a teacher model.

By prompting GPT-4 with the task description and section description, we ask it to generate the contents of the section. To ensure that the generated section-wise prompts are concise and relevant, we prompt GPT-4 to not generate more than five lines of content for each section. We use LLAMA2-13B model, which we finetune using LoRA adapters as the auxiliary LM that generates sections.

### A.4 Prompt to Llama2-13B for fine-tuning

```
### Instruction:
You are a prompt engineer, you have to write a structured prompt.
For the given task description, examples and section description,
```

write the contents of the section that align with
section description.

### Task Description:
{data_point['task_description']}

### Section Description:
{data_point['section']}:{section_descriptions[data_point['section']]}

### Response:
{data_point['prompt']}

### A.4.1 EXAMPLES OF TRAINING DATASET

Task: super glue boolq
Task Description: With no explanation, label the following with
either False or True.
Section: description
Prompt:  As a prompt engineer, your task is to create a prompt
that accurately labels statements as either True or False
without providing any explanations. The statements may involve
general knowledge, trivia, or facts about events, places,
people, or objects. Ensure you have a strong understanding of
the subject matter and write a concise and specific task
description related to the given statements.

======================================================

Task: glue qnli

Task Description: With no explanation, label A to B with either
entailment or not entailment

Section: background

Prompt: 1. Entailment means that the information in statement B
can be inferred directly from statement A.

2. Not entailment means that the information in statement B
cannot be inferred directly from statement A or is unrelated.

3. Understanding the context and relationship between the two
statements is crucial for accurate classification.

======================================================

Task: bigbench undo permutation

Task Description: In the following sentences with shuffled
words, find the correct order of word swaps to unscramble the
sentence.

Section: tricks

Prompt: 1. Identify the key words or phrases in the task to
understand the context of the sentence. Look for nouns, verbs,
and adjectives that seem related or could logically fit together.

2. Start by solving the problem step by step and focus on one

swap at a time. Breaking the problem into smaller sub-problems
will make it easier to manage.

3. To make the task more manageable, first focus on swapping the
words that are clearly out of place, such as words that should
be at the beginning or end of the sentence.

## A.5 PROMPT FOR IDENTIFYING IMPORTANT FACETS

you are given a task, along with it's description, some examples
of how to solve the task and section descriptions.
What do you think would be the most important sections to
include for the given task.
## Task
{task}
## Task Descirption
{tas_description}
## Examples
{Examples_string}
## Section Descriptions
{sections}

## A.6 CLUSTERING TYPE 1

You are given a science question, you need to tell which broad
topic is this question from.
Question: {train_questions_new[ij]}
Answer: {answer}
Give your answer as a single word, between <Answer></Answer>
tags like: <Answer>Thermodynmics</Answer> or
<Answer>Botany</Answer>.
Subtopic:

## A.7 CLUSTERING TYPE 2

You are given a set of feedbacks, you need to cluster them into
five groups based on similarity, and then provide a summary of
each group. You can use the following feedbacks to cluster: \n
{feedback}

provide each cluster explnation within the following tags:
<Cluster></Cluster>

You are given a feedback and a set of clusters, you need to tell
which cluster this feedback belongs to.

The clusters are: \n {string_of_clusters}

The feedback is: {feedback}

give your final answer as the number of the correct cluster
between <Answer></Answer> tags like: <Answer>1</Answer>.'''

## A.8 FEEDBACK PROMPTS

**Feedback over mini-batch**

```
You are a teacher and you have to give feedback to your
students on their answers.

You are teaching how to solve math problems to your students.
You are given a question, it's true answer and answer given by
student. You are also given the explanations written by your
students while solving the questions.

The questions are answered wrong by the students.
You have to tell why is the solution wrong and what information
is can be added to the in the Background Knowledge part that
would have helped the student to write better explanations.

## IMPORTANT: You are also given a history of changes you made
to the background knowledge part and the change in student's
accuracy after making the change. You have to use this history
to make your feedback.

Be explicit and tell the exact information that can be added
without further modification / addition.

### IMPORTANT: Give feedback in form of instructions like  add a
section, add a subsection, set the content of a section, set the
content of a subsection, delete a section or delete a subsection
in the background knowledge part.

Give very granular feedbacks, like if the student has made a
mistake in the calculation, then tell what is the mistake in the
calculation and how to correct it, if the student has made a
mistake in the concept, then tell what is the mistake in the
concept and how to correct it.

## Background Knowledge
    {current_prompt}

## History
    {history_string}

Now, it is your turn to give feedbacks to the students.
You can only provide a one line feedback.

=======================================
```

**Feedback over batch**

```
You are given a set of feedbacks for some problems. The set
feedbacks for each problem separated by ========== symbol.
You have to summarize the feedbacks into a final feedback.
You are also given a set of wrong questions. You need to tell
which edit can be applied to aid the student in solving the
wrong question.

To achieve your task, try to follow the following steps;
1. Identify the general problem that is being solved by all the
feedbacks.
2. Once you have identified the problem, try to make a new
```

feedback that covers most of the
feedbacks given.
Let's say the problem in the first feedback is the absence of
methods to solve linear equation and in the second feedback it
is the method to inverse a matrix.
You know that both of these problems can be caused by adding how
to solve convert a matrix  into row rediced echolon form. So,
add that.
3. Try and validate your feedback. Once, you have a feedback try
to see if it covers every
feedback, if it does not cover any feedback, add that to your
new feedback.
4. See the wrong questions and try to identify what is the
problem in the question.
If the problem is not covered by your feedback, add that to your
feedback.
5. You can add specifics like examples, definitions etc make
sure that the feedback is enough to be directly added without
any modification.

You may use the following function templates-

add_section(sectioname)
add_subsection(section_name, subsection_name)
set_section_content(section_name, new_content)
set_subsection_content(section_name, subsection_name, new_content)
delete_section(section_name)
delete_subsection(section_name, subsection_name)

Your summary cannot include more than four functions.
Make sure that the content is useful,
not just a very general statement. Something specific.

Instructions:
{edits}

Wrong Questions:
{wrong_examples_string}

Summary:

## A.9 EDITING PROMPT

You are given an input prompt and a feedback, you have to
incorporate the feedback into the input prompt and output the
final prompt.
An example of the task is given below

### Input Prompt
Introduction: In this task you have to answer the given question.

### Feedback
The background knowledge is incomplete, it does not include what
are the factors that affect the water usage and how many water
sources are there.
\\add_subsection("Background Knowledge")
\\add_subsection_content(water usage depends on the population,
climate, economic development, and availability of water
sources. There are two sources of water, surface water and

```
groundwater.)

### Final Prompt
Introduction: In this task you have to answer the given question.
Background Knowledge: water usage depends on the population,
climate, economic development, and availability of water
sources. There are two sources of water, surface water and
groundwater.

Only output the final prompt nothing else.

### INPUT PROMPT
{current_prompt}

### FEEDBACK
{edits}

### FINAL PROMPT
```

## A.10 EXAMPLE OF PROMPT EVOLUTION USING OUR METHOD

See example in Figure 3.

---

**Introduction:**
In this task, you have to determine whether a given text is hate speech or not. 0 means Non-Hate and 1 means Hate.

---

**Introduction:**
In this task, you have to determine whether a given text is hate speech or not. 0 means Non-Hate and 1 means Hate.

**Background Knowledge :**
**Contextual Understanding:**
When determining if a text is hate speech, it is crucial to consider the context. Not all negative or critical statements are hate speech. Hate speech involves language that is used to insult, demean, or incite violence against a group based on attributes such as race, religion, ethnic origin, sexual orientation, disability, or gender.

**Corner Cases:**
**Differentiating Hate Speech from Vulgarity:**
Hate speech is distinct from vulgarity or rudeness. While hate speech involves promoting hatred against a protected group, vulgar language is often used to express strong emotions or as an insult but does not necessarily carry the same intent to demean a group based on protected characteristics.
**Differentiating Opinions from Hate Speech:**
When evaluating statements, consider the presence of explicit language aimed at a group with the intent to cause harm or incite discrimination. Opinions, even if controversial or unpopular, do not automatically qualify as hate speech unless they contain elements that specifically target a group with hateful intent

---

**Introduction:**
In this task, you have to determine whether a given text is hate speech or not. 0 means Non-Hate and 1 means Hate.

**Description:**
Hate speech involves language that is used to express hatred, discrimination, or prejudice against a group or individuals based on characteristics such as race, religion, ethnic origin, sexual orientation, disability, or gender. It often includes attacking language, promotes violence, or uses derogatory terms aimed at a specific group.

**Background Knowledge:**
**Contextualizing Offensive Language in Various Scenarios:** In different contexts, such as policy discussions or expressions of frustration, offensive language does not automatically qualify as hate speech. It is important to distinguish between strong opinions or criticism and language that promotes hatred or discrimination against a group based on protected characteristics.
**Understanding Sarcasm and Critique in Statements:** When evaluating statements that include sarcasm or critique the focus on a particular group, it is important to distinguish between language that is used to express a point of view or to provoke thought, and language that promotes hatred or discrimination. Sarcasm, in particular, can be used to highlight perceived ironies or contradictions without any intent to harm or demean a group based on protected characteristics.

**Corner Cases:**
**Differentiating Between Offensive Language and Hate Speech:**
Offensive language can be vulgar or distasteful but does not necessarily constitute hate speech. Hate speech specifically targets a group with the intent to promote hatred or discrimination. Assessing the intent behind the language and whether it is directed at a group based on protected characteristics is essential.
**Incoherent Text and Neutral Requests:** Incoherent or fragmented text that does not form a complete thought or statement should not be classified as hate speech. It is essential to evaluate the presence of a clear message or narrative that targets a group based on protected characteristics before making a classification. Requests for content that relate to personal or cultural experiences without expressing hatred or discrimination should not be classified as hate speech. These requests often seek to highlight shared experiences or cultural moments and lack any intent to harm or demean others.

**Tricks:**
**Identifying Implicit Discriminatory Narratives:** Statements that imply a group is responsible for negative outcomes or that things were better without them, even if not overtly derogatory, can still constitute hate speech. Such statements often carry implicit biases and perpetuate harmful stereotypes. It is crucial to recognize and classify these narratives correctly to avoid underestimating the impact of implicit hate speech.
**Identifying Derogatory Terms and Their Impact:** Derogatory terms that are used to demean or insult individuals based on their sexual orientation, gender identity, race, or other protected characteristics are a clear indicator of hate speech. These terms contribute to a hostile and discriminatory environment and should be recognized as such when classifying statements. Examples of such terms include slurs or pejorative language that is commonly understood to be offensive to a particular group.

Figure 3: Evolution of prompts through iterations of UNIPROMPT on the Ethos dataset. Starting from a simple one-line prompt having an accuracy of $82\%$, UNIPROMPT adds background knowledge, corner cases, and additional sub-sections yielding a prompt with accuracy $88\%$. After further iterations, our algorithm converges to a detailed, human-like longform prompt that achieves accuracy of $92\%$.

Table 7: Comparison with ALPE method that produces long prompts on BigBenchHard.

| Dataset | Original Prompt | ALPE | UNIPROMPT |
|---|---|---|---|
| Causal Judgement | 58.9 | 63.7 | **64.5** |
| Formal Fallacies | 60.0 | 73.1 | **77.5** |
| Hyperbation | 74.4 | 85.5 | **89.8** |
| Logical Five | 38.8 | 54.7 | **58.5** |

### A.11 COMPARISION OF OUR METHOD WITH EXISTING METHODS

See Figure 4.

```
Human Prompt
Let's differentiate using step by step reasoning like a medical
expert.
Our Prompt
Introduction:  In this task, you are given a medical question.  You
have to solve the question.
Description:  To solve medical questions effectively, it is
important to understand various medical conditions, their
progression, and associated clinical features.
Background Knowledge:  Differential Diagnosis of Subcutaneous
Nodules:
When evaluating subcutaneous nodules, consider mobility,
consistency, and skin adherence.  Epidermoid cysts are firm,
non-tender, and the skin cannot be pinched over them.  Lipomas are
soft, mobile, and have pinchable skin.
Corner Cases:  Antiretroviral Therapy Complications:
Doctor should be aware of the common side effects of antiretroviral
drugs, with specific attention to the association between
didanosine and pancreatitis, and the recommended management
strategies, such as replacing didanosine with lamivudine.
```

Figure 4: Comparison of human-written Prompt and prompt produced by UNIPROMPT on MedQA dataset.

### A.12 COMPARISON WITH LONG PROMPT ENGINEERING METHOD OF HSIEH ET AL. (2023)

Generation of long, sectioned prompts with facets required to solve the task is a technical contribution of our method. As mentioned in Related Work, Hsieh et al. (2023) use greedy and genetic algorithms to edit lengthy prompts. We now present comparisons with the ALPE method of Hsieh et al. (2023) on the BigBenchHard datasets they evaluate on. The results are given in Table 7. UNIPROMPT consistently outperforms ALPE on the four datasets.

### A.13 EFFECT OF LENGTH ON PERFORMANCE OF PROMPT

Here we answer the question: *How much does only length contribute to* UNIPROMPT*'s success?*. To answer this, we replace the prompt with in-context examples of the same context length and compare the accuracies in Table 8. We also compare the case where we include only the examples that the solver LLM gives incorrect prediction on, denoted as "Wrong ICL" row in the table. We see that there is a slight increase in accuracy when wrong examples are included in the prompt over randomly including examples. But, overall, UNIPROMPT performs much better than including in-context examples. This shows that length is not the only factor contributing to UNIPROMPT's success.

**OPRO optimized prompt**
```
Start by dissecting the problem to highlight important numbers and
their relations.  Decide on the necessary mathematical operations
like addition, subtraction, multiplication, or division, required
for resolution.  Implement these operations, keeping in mind any
units or conditions.  Round off by ensuring your solution fits the
context of the problem to ensure accuracy
```
**Our Prompt**
```
Introduction:  In this task, you are given a math question.  You
have to solve the question.
```
**Strategies for Word Problems:**
```
1.  Understanding Word Problems:  When solving word problems, it
is crucial to read each sentence carefully and comprehend the time
periods and quantities involved.  Avoid incorrect multiplication
or addition by paying close attention to whether a quantity remains
constant over a period or changes.  If a quantity is consistent,
it does not need to be multiplied by the number of days or weeks
unless the problem specifies otherwise.
2.  Calculating Averages:  To calculate the average of a set of
numbers, add all the numbers together and then divide by the number
of items.  In word problems, ensure you have the correct total
before dividing by the number of periods, such as weeks, to find
the average for each period.
3.  Understanding Past and Future Events in Word Problems:
Distinguish between past and future events by identifying the
starting and ending points.  To calculate the time interval between
two events, determine the direction of time from past to future and
compute the interval accordingly.  This understanding is essential
when dealing with problems that ask for the time since a past event
or until a future event.
```

Figure 5: Comparison of prompt produced by the state-of-the-art ORPO (Yang et al., 2023) and by UNIPROMPT on the GSM8K dataset.

Table 8: Analysis of the effect of length and contents on the performance of UNIPROMPT

|  | Ethos | ARC | GSM8K |
| --- | --- | --- | --- |
| UNIPROMPT | 93.7 | 90.5 | 82.4 |
| ICL Prompt | 63.0 | 86.7 | 76.3 |
| Wrong ICL | 70.4 | 87.1 | 78.2 |
| Summarized Prompt | 84.3 | 85.5 | 66.0 |

### A.14 DO DIVERSE TASK FACETS ORGANIZED AS SECTIONS REALLY HELP?

We want to empirically validate if all the diverse task facets that UNIPROMPT learns indeed contribute to the performance gains that we observe in Table 1. We consider two ablations:

**1)** We successively remove each facet (i.e., sections) in the learnt prompt for the task and report the performances of the prompts with fewer facets. In Figure 6, for the Ethos dataset, we see that almost every additional facet contributes to non-trivial gains in accuracy.

**2)** Could we have captured the information differently and retained the performance? We do a simple experiment – we summarize all the facets (i.e., learnt prompt) and evaluate the resulting prompt. In Figure 6 (right) (green line), we see that the summarized prompt has a significant accuracy drop.

### A.15 SENSITIVITY TO PROMPTS USED FOR EXPERT LLMS IN UNIPROMPT

The prompts used for expert LLMs in our algorithm, i.e., for clustering, feedback over batches and mini-batches, and editing, do matter for obtaining good performance. However, note that the prompts are task-agnostic and can be used as-is for new tasks. Moreover, prompts for clustering and editing

Table 9: Sensitivity of UNIPROMPT to expert LLM prompts, on the Ethos dataset.

| Expert LLM Prompt for UNIPROMPT | Test Accuracy |
|---|---|
| Simple prompt for mini-batch feedback | 83.5 |
| Simple prompt for batch feedback | 91.0 |
| Detailed prompts (Appendix A.8) | 93.7 |

are very simple and involved minimal human effort. Further, to study the reliance of UNIPROMPT on the quality of feedback prompts, we run an ablation study, where we replace the engineered prompts for feedback at batch and mini-batch levels with simpler prompts. The results are given in Table 9 for the Ethos dataset. We observe that the performance of UNIPROMPT depends heavily on the prompt used for obtaining feedback at mini-batch level; whereas simplifying prompt for feedback at the batch level has much less impact on the final accuracy.

### A.16    UNIPROMPT TRAINING BEHAVIOR

An example of evolution of prompts using our algorithm is given in Appendix 3. It starts with a simple description of task and adds important facets like *differentiating between hate speech and rudeness*. In contrast, **ProTeGi** (Pryzant et al., 2023) yields a rather terse prompt on the same dataset: "*Does the following text contain language that targets a group of people based on their religion, gender, or other personal characteristics?*".

The training curves in Figure 6 show that our method initially performs edits on the prompt that simultaneously increase the train as well as the validation accuracy. After about 10 or 15 iterations (each batch update is an iteration), validation accuracy decreases while train accuracy continues increasing, indicating overfitting; which we overcome using early stopping.

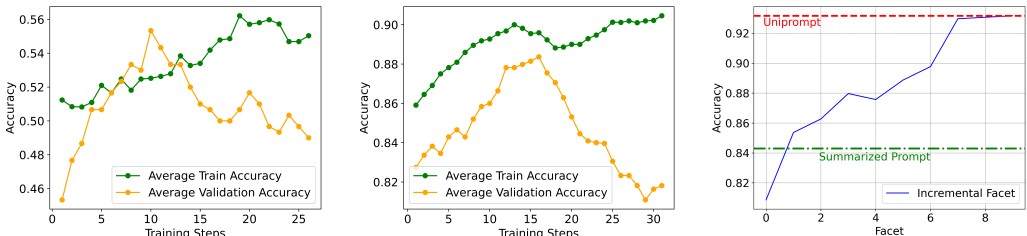

Figure 6: Training curves for MedQA (left) and ARC (middle) datasets when UNIPROMPT is initialized with (published) state-of-the-art prompts; (right) ablation of facets on Ethos.

