# OpenReview forum: "Task Facet Learning: A Structured Approach to Prompt Optimization"
_ICLR.cc/2025/Conference — Submitted to ICLR 2025_

### Official Review · Reviewer_veyq · 2024-10-20

**Soundness:** 3
**Presentation:** 3
**Contribution:** 2
**Rating:** 5
**Confidence:** 4

**Summary:**

This paper presents an improvement to current prompt optimization approach. It does so by clustering the input space and enabling the optimization algorithm to learn multiple facets of a task from a set of batched clusters. The paper discusses the empirical observation that inspires the improvement and conducts experiments to demonstrate its efficacy. Results suggest that the framework achieve better results than existent approach.

**Strengths:**

1. The paper is well-structured and the story is well-told. It starts with essential observations that motivate the improvements so that the approach bears heuristic justification.

2. Experiments are comprehensive, including several baselines, and benchmarks of different types. Experiment results also provide strong support to the thesis of the paper.

**Weaknesses:**

My concern on the improvement is its generalization ability. The experiments cover hate speech detection, math, QA but there are many other task types not covered in this dataset. For example, OPRO and [Li et al 2024](https://arxiv.org/abs/2409.15199) ran experiments on BBH which contains a much more diverse task portfolio than presented in the paper. Since the proposed improvement centers around clustering the training examples, whether the samples can be cluster into meaningful subspace has a direct impact on optimization performance. But we don't see sufficient evidence to demonstrate the approach works on a divers task portfolio.

Besides, the prompt optimization is very sensitive to the initial prompt and iterations (also discussed in Sec 2.1), but the reported metrics don't take the performance variation into consideration. I have concern on the robustness of the results.

**Questions:**

1. If I understand correctly, the wrongfully predicted training samples are clustered only once at onset, how does the algorithm handle the wrong samples after the optimization starts? Does it use them to generate feedback?

2. Did you try using other model families? How does the performance look like when the expert model is not as powerful as GPTs?

3. The clustering is conducted through LLM, but this could be an issue when the number of examples increases. Did you try other clustering approach? which might make it feasible to dynamically cluster the examples along the iterations.

---

> ### Author Response · Authors · 2024-11-21
>
> Thank you for reading the paper carefully, for the helpful feedback and the questions.
>
> > My concern on the improvement is its generalization ability. The experiments cover hate speech detection, math, QA but there are many other task types not covered in this dataset. For example, OPRO and Li et al 2024 ran experiments on BBH which contains a much more diverse task portfolio than presented in the paper. Since the proposed improvement centers around clustering the training examples, whether the samples can be cluster into meaningful subspace has a direct impact on optimization performance. But we don't see sufficient evidence to demonstrate the approach works on a divers task portfolio.
>
> Beyond Table 1, we experiment on other tasks too (see Tables 2 and 4):
> * code generation (HumanEval, MBPP, Table 2)
> * four diverse medical QA tasks (PubmedQA, MedMCQA, MedQA, MMLU MG, Table 4).
>
> **BBH-Hard.** Below we present comparisons to OPRO on a subset of BBH tasks. We don’t include Li et al 2024 which uses a much more powerful solver model (Claude-3 Sonnet), whereas we use GPT-3.5, due to time & compute constraints. For fair comparison, we evaluate our method on the same task-wise test sets as OPRO. For OPRO, we evaluate their published best prompts on the tasks on the respective test sets. We observe that on 3/4 datasets, UniPrompt gets good to significant gains compared to OPRO. We use one-line task description as the init prompt for UniPrompt (column 2 in the Table below).
>
> | Task      | Init Prompt | OPRO      |  UniPrompt(Ours)      |
> | ------------- | ------------- | ------------- | ------------- |
> | Causal Judgement | 54.29% | 57.24% |  **59.37%** |
> | Movie Recommendation | 58.04% | **77.81%** | 71.80% |
> | Snarks | 67.00% | 67.88% |  **74.30%** |
> | Sailent Translation Error Detection | 42.59% | 50.61% | **50.77%** |
>
> On the Move Recommedation dataset, here are the prompts learnt by OPRO vs UniPrompt. It is evident that UniPrompt's prompt is much more informative compared to that of OPRO. However, on the small test set of 200 examples, OPRO's prompt seems to perform much better.
>
> **OPRO** _Let’s uncover the perfect movie recommendation from the options provided, ensuring an exceptional cinematic experience together as we select the most captivating and satisfying choice that will keep us thoroughly engaged and immersed until the very end._
>
> **UniPrompt** _Given a set of movies and five options, pick the movie that most closely matches with the given set of movies in the statement. Strictly output (A), (B), (C), (D), or (E)._
>
> _Labelling Guidelines: When selecting a movie, ensure that it aligns with the overarching themes, genres, narrative style, and tone of the majority of the movies in the given set. Avoid focusing too narrowly on matching just one movie's theme or genre; instead, consider the overall balance and diversity of these elements in the set. Additionally, use a holistic approach that includes thematic and genre alignment, as well as the overall tone and critical acclaim of the movies in the given set. This ensures a well-rounded decision that encompasses the broadest range of characteristics present across all movies._

---

> > ### Comment · Reviewer_veyq · 2024-11-24
> >
> > Thanks for the response. How are the four BBH tasks selected? For the generalization capability, why don't you also compare your approach with baselines on some other benchmarks you experimented on (e.g., the ones in Table 2 & 4)? What is the rationale behind the decision?

---

> > > ### Author Response · Authors · 2024-11-24
> > >
> > > Thank you for the response.
> > >
> > > To start with, we chose these four tasks because they are among the tasks with low reported accuracies. We are continuing to experiment with more tasks and will share the updated results in a few days.
> > >
> > > Thanks for the suggestion on adding more baselines for Table 2 and 4. In table 4 we had included MedPrompt which is the state-of-the-art with GPT-4 on medical QnA tasks, but we will conduct experiments with more baselines and report back.

---

> ### Author Response · Authors · 2024-11-21
>
> > Besides, the prompt optimization is very sensitive to the initial prompt and iterations (also discussed in Sec 2.1), but the reported metrics don't take the performance variation into consideration. I have concern on the robustness of the results.
>
> We agree that prompt optimization can be sensitive to the **initial prompt**. Therefore, to study the robustness of UniPrompt to the initial prompt, we considered three types of initializations: Task description, Expert Prompt, Llama prompt in our paper. Table 5 (appendix) shows the accuracy of UniPrompt when initialized with these three kinds of prompts, besides the main results in Table 1 with the task description initialization. We reproduce a combined version of Tables 1 and 5 below for clarity.
>
>
> **Ethos**
> | Init Prompt | Baseline |  UniPrompt(Ours)      |
> | ------------- | ------------- | ------------- |
> | Expert Prompt | 74.1% |  **84.0%** |
> | Llama Prompt | 74.0% | **92.0%** |
> | Task Description | 76.8% |  **92.3%** |
>
> **ARC**
> | Init Prompt | Baseline |  UniPrompt(Ours)      |
> | ------------- | ------------- | ------------- |
> | Expert Prompt | 78.4% |  **86.0%** |
> | Llama Prompt | 89.7% | **90.5%** |
> | Task Description | 79.7% |  **86.0%** |
>
> **MedQA**
> | Init Prompt | Baseline |  UniPrompt(Ours)      |
> | ------------- | ------------- | ------------- |
> | Expert Prompt | **53.1%** |  52.3% |
> | Llama Prompt | 52.6% | **55.5%** |
> | Task Description | 52.7% |  **57.1%** |
>
> **GSM8K**
> | Init Prompt | Baseline |  UniPrompt(Ours)      |
> | ------------- | ------------- | ------------- |
> | Expert Prompt | 78.9% |  **82.4%** |
> | Llama Prompt | 79.5% | **81.5%** |
> | Task Description | 59.4% |  **82.4%** |
>
> Across all prompt initializations, we see a significant increase in accuracy using UniPrompt (except for one case in MedQA). Notably, in all cases, starting from a simple one-line task description, UniPrompt is able to reach accuracies comparable to or better than what it could starting from the expert prompt.
>
> As for **iterations**, we did observe an overfitting trend as the number of epochs increased --- train error keeps decreasing while the validation error may increase. Therefore, we employ an early stopping criterion based on validation accuracy, as mentioned in Section 4.
>
>
> > 1. If I understand correctly, the wrongfully predicted training samples are clustered only once at onset, how does the algorithm handle the wrong samples after the optimization starts? Does it use them to generate feedback?
>
>
>
> Yes, the wrongfully predicted samples are clustered only once in the beginning, and yes, they are used to generate feedback. Specifically, UniPrompt creates batches such that all samples in a given batch belong to the same cluster. It then obtains feedback from the Expert LLM on how to update the prompt such that the solver LLM’s (mis)predictions can be improved (see Section 3.1.2).
>
>
>
> > 2. Did you try using other model families? How does the performance look like when the expert model is not as powerful as GPTs?
>
> We want the expert model to be powerful to generate various kinds of feedback throughout the algorithm. That said, we do provide some promising results in Table 3, where we see that GPT-3.5 can also be a good expert model, compared to GPT-4.
>
>
>
> Note that we expect prompt optimization to be a one-time procedure, after which the solver model is deployed to serve millions of requests (for example, see the real-world case study in Sec. 4.4). For efficiency purposes, it is important to ensure that the solver model is an efficient one. Therefore, an alternative ablation is if UniPrompt can improve the performance of less powerful models such as Llama-8B or Llama-70B when they are used as the solver model. We share these results in our responses to Reviewer SD4J. We summarize it below for convenience.
>
>
>
> On the ARC-challenge dataset, UniPrompt with Llama3 8B Instruct as the solver model yields 91.67% accuracy (using one-line task description as the Init prompt) compared to the baseline performance reported in their model card which is 82.4%. Similarly, for Llama3 70B Instruct as the solver model, UniPrompt achieves 95.58% accuracy as against the baseline accuracy of 94.4% reported on the model card.
>
> > 3. The clustering is conducted through LLM, but this could be an issue when the number of examples increases. Did you try other clustering approach? which might make it feasible to dynamically cluster the examples along the iterations.
>
> Great point. Yes, re-clustering the samples every few epochs can be a useful addition. We tried clustering using smaller models such as Llama 3.1 70B but they were not effective. As the reviewer suggests, we are currently trying embedding-based approaches as alternatives.

---

> ### Author Response · Authors · 2024-11-26
>
> >  why don't you also compare your approach with baselines on some other benchmarks you experimented on (e.g., the ones in Table 2 & 4)? What is the rationale behind the decision?
>
> **Table 4** contains the comparison between UniPrompt with Medprompt which is the state-of-the-art with GPT-4 on medical QnA tasks. Based on the reviewer's suggestion, below we present the comparison of UniPrompt with other baselines.
>
> | Method      | MedQA | PubMedQA      | MedMCQA      | MMLUMG |
> | ------------- | ------------- | ------------- | ------------- | ------------- |
> | MI(Manual Instructions)* | 77.83% | - | 65.87% | - |
> | COT* [(Wei et al., 2022)](https://arxiv.org/pdf/2201.11903) | 49.10% | - | 59.07% | - |
> | APO*[(Pryzant et al., 2023)](https://arxiv.org/pdf/2305.03495) | 77.41% | - | 65.93% | -|
> | OPRO*[(Yang et al., 2023)](https://arxiv.org/pdf/2309.03409) | 76.56% | - | 66.00% | - |
> | EvoPrompt*[(Guo et al., 2023)](https://arxiv.org/pdf/2309.08532) | 77.15% | - | 65.47% | - |
> | STRAGO* [(Wu et al., 2024)](https://arxiv.org/pdf/2410.08601v1) | 80.05% |- | 67.20% | - |
> | Ours | **80.9%** | 70.3% | **79.2%** | 78.0% |
> | **Methods with few-shot examples and/or multiple LLM inference calls** | | | | |
> | Ours+kNN | 81.0% | 72.2% | 81.4% | 94.0% |
> | Ours+kNN+CoT | 83.9% | 74.7% | 82.6% | 96.0% |
> | Ours+kNN+CoT+Ensemble | **87.0%** | **75.6%** | **84.5%** | **99.0%** |
> | MedPrompt [(Nori et al., 2023)](https://arxiv.org/pdf/2311.16452) | 80.6% | 71.2% | 79.1% | 98.0% |
>
> *Indicates the results from [Wu et al. (2024)](https://arxiv.org/pdf/2410.08601v1). Results on PubMedQA and MMLUMG not provided.
>
> **Table 2** presents the performance of UniPrompt on code generation datasets. Although we intended to include accuracy results for additional baselines, most  prompt optimization baselines (e.g., from Table 1)  do not conduct experiments on code generation tasks. So, we compare to a recent prompt optimization study ([Wang et al. (2024)](https://arxiv.org/pdf/2409.16416)) focused on code generation. The best method reported in their paper achieves 85.4% on HumanEval and 68.2 on MBPP using GPT-4o, significantly lower than UniPrompt's performance with GPT-4-turbo (93.8 and 92.5 respectively).
>
> ----------------------------------------------------------------
>
> **Additional Results on BBH (10 datasets)**.
> We selected 10 datasets, spanning the **four categories** defined by Li et al 2024:
>
> 1. Algorithmic and Multi-Step Arithmetic Reasoning (Boolean Expression, Logical Deduction five objects, Navigate)
> 2. Natural Language Understanding (Snarks, Salient Translation Error Detection, Disambiguation_QA, Formal Fallacies)
> 3. Use of World Knowledge (Causal Judgement, Movie Recommendation, Date Understanding), and
> 4. Multilingual Knowledge and Reasoning (Salient Translation Error Detection).
>
> We compared the performance of UniPrompt(Ours) with OPRO on these datasets. The results are shown in the table below.
>
> | S.No | Task                                     | Init Prompt | OPRO    | UniPrompt(Ours) |
> |------|------------------------------------------|-------------|---------|---------|
> | | **Algorithmic and Multi-Step Arithmetic Reasoning** | | | |
> | 1    | Boolean Expression   | 83.64%    | 78.74%  | **92.37%**  |
> | 2 | Logical Deduction five objects | 29.53%  | 38.97% | **39.62%** |
> | 3   | Navigate  | 60.95%   | 51.74%  | **77.16%**  |
> | | **Natural Language Understanding** | | | |
> | 4    | Snarks | 67.00%      | 67.88%  | **74.30%**  |
> | 5    | Disambiguation_QA | 53.30%  | 57.43%  | **67.05%**  |
> | 6    | Formal Fallacies | 57.60% | 53.14%  | **57.90%**  |
> | | **Use of World Knowledge** | | | |
> | 7    | Causal Judgement | 54.29%  | 57.24%  | **59.37%**  |
> | 8    | Movie Recommendation | 58.04%   | **77.81%**  | 71.80%  |
> | 9    | Date Understanding | 74.21%  | 52.59%  | **81.96%**  |
> | | **Multilingual Knowledge and Reasoning** | | | |
> | 10    | Salient Translation Error Detection | 42.59%  | 50.61%  | **50.77%**  |
>
> From the table, it is evident that UniPrompt (Ours) shows a significant improvement over OPRO for a majority of tasks. For instance, it achieves significantly higher accuracy in tasks such as "_Boolean Expression_" (**92.37%** vs. 78.74%), "_Date Understanding_" (**81.96%** vs. 52.59%), and "_Navigate_" (**77.16%** vs. 51.74%). However, there is one exception: in the "_Movie Recommendation_" task where OPRO achieves a higher accuracy (77.81% vs. 71.80%). As mentioned in our previous response, it is evident that UniPrompt's prompt is much more informative compared to that of OPRO. However, on the small test set of 200 examples, OPRO's prompt performs better.

---

> > ### Author Response · Authors · 2024-11-29
> > **Request for feedback on our response**
> >
> > Dear Reviewer veyq,
> >
> > Thank you once again for your valuable feedback and constructive discussion. We have included additional experiments on the BBH dataset and added new baselines for MedQA.
> >
> > We hope these new experiments address your concerns. Please let us know if you have any further questions.

---

> > > ### Comment · Reviewer_veyq · 2024-12-02
> > >
> > > Thanks for the response. I will keep my score at this moment.

---

> > > > ### Author Response · Authors · 2024-12-03
> > > > **Thanks**
> > > >
> > > > Dear reviewer, Given that the discussion window is closing soon, it would be very helpful if you could let us know why you are still on the fence about the paper. It would help us prepare the next version better, in the worst case. Thanks again, for your time and feedback!

---

### Official Review · Reviewer_Ynwp · 2024-10-26

**Soundness:** 3
**Presentation:** 3
**Contribution:** 3
**Rating:** 6
**Confidence:** 4

**Summary:**

This paper introduced a method for prompt optimization called UNIPROMPT. The prompt optimizaiton is deemed as the task of learning multiple facets of a task. UNIPROMPT exploit the structure in the prompt optimization and use a clustering approach to group examples with similar task facets. Experimental results on multiple datasets demonstrate that prompts generated using UNIPROMPT achieve higher accuracy than human-tuned prompts and state-of-the-art methods.

**Strengths:**

(1) The paper introduces a new method for prompt optimization and consider multiple facets of a task and leveraging the structure in the prompt.
(2) The proposed clustering approach and two-tier feedback mechanism provide a systematic way to generate prompt edits that capture generalizable concepts.
(3) The experimental results on several datasets shows that UNIPROMPT outperforms human-tuned prompts and SOTA methods.

**Weaknesses:**

(1) I am skeptical about the current importance of prompt optimization. Prompt optimization is more of a need in an era when large models are not powerful (for example last year). Nowadays, the instruction model follows human instructions very well, and many of them are 0-shot, see llama-3.1 model card. I would like to hear the author's opinion on this.

(2) For the results in table 1, it is recommended to add an average column to make the presentation clearer.

(3) I found that on the ARC task, compared with the llama prompt, the optimized prompt led to a decrease in performance. Is there any explanation?

**Questions:**

N/A

---

> ### Author Response · Authors · 2024-11-21
>
> Thank you for reading the paper carefully, for the helpful feedback and the questions.
>
> > (1) I am skeptical about the current importance of prompt optimization. Prompt optimization is more of a need in an era when large models are not powerful (for example last year). Nowadays, the instruction model follows human instructions very well, and many of them are 0-shot, see llama-3.1 model card. I would like to hear the author's opinion on this.
>
> While models continue to improve in their instruction-following capabilities, we see significant improvements in task accuracy even for the latest models such as LLama-3.1 and GPT-4.  For example, on the LLama 3.1 model, applying UniPrompt increases accuracy on ARC-challenge dataset to 91.67%, compared to [82%](https://github.com/meta-llama/llama-models/blob/main/models/llama3_1/MODEL_CARD.md) reported in its model card. And from Tables 2, 3, and 4, we see that we can significantly improve the performance of state-of-the-art GPT-4 using prompt optimization across several datasets. Thus, we believe prompt optimization is important even in the era of increasingly powerful LMs.
>
> More generally, as general-purpose LMs are becoming the go-to option even for specialized and challenging tasks (such as the search query intent match problem we discuss in Section 4.4), it is all the more important to help practitioners get the best possible accuracy. In these settings, the (only) knob at their disposal is the model prompt. Therefore, prompt optimization is still very relevant in practice.
>
> Also, while prompt-based accuracy gains are useful for end-tasks, they are even more critical when the LLM is used as a verifier/evaluator for a target model or is used to distill a smaller model -- since the accuracy gains directly translate to the resultant model quality.
>
> > (2) For the results in table 1, it is recommended to add an average column to make the presentation clearer
>
> Thanks, we will make this addition in the revised version.
>
> > (3) I found that on the ARC task, compared with the llama prompt, the optimized prompt led to a decrease in performance. Is there any explanation?
>
> Yes, this is due to a difference in the initialization prompt. Please refer to Table 5 where we report results of UniPrompt when initialized with the Llama prompt. The optimized prompt obtains 90.5% accuracy, higher than the initial Lllama prompt (89.7%).
>
> In comparison, Table 1 reports UniPrompt’s results when initialized with the one-line Task Description. Here the optimized prompt’s accuracy (86%) is significantly higher than the Task Description initialization (79.7). With a greedy selection, the accuracy of UniPrompt increases further.

---

> ### Author Response · Authors · 2024-11-25
> **Request for Feedback on the rebuttal**
>
> Dear Reviewer **Ynwp**,
>
> Thank you again for your helpful feedback. We’ve done our best to address your concerns with additional experiments.
>
> Please let us know if you have any further questions or comments after reading our rebuttal.

---

> > ### Comment · Reviewer_Ynwp · 2024-11-28
> >
> > Thanks for the authors' response. I will keep my positive rating.

---

### Official Review · Reviewer_SD4J · 2024-11-01

**Soundness:** 3
**Presentation:** 3
**Contribution:** 3
**Rating:** 5
**Confidence:** 3

**Summary:**

The paper introduces UNIPROMPT, a method that uses task facet learning to enhance prompt optimization by breaking down prompts into distinct semantic sections, targeting diverse facets such as counter-examples and explanations.

**Strengths:**

- UNIPROMPT clusters examples to capture task-specific facets effectively, with a two-tier feedback mechanism ensuring that prompt modifications are generalizable and not overly specific to individual examples.
- The authors provides strong empirical results, with UNIPROMPT outperforming state-of-the-art methods on multiple datasets, notably achieving significant improvements on hate speech classification.
- By focusing on automatic prompt generation without relying heavily on human-engineered prompts, this approach may reduce the manual overhead of prompt tuning.
- Extensive ablations validate the contribution of each component (clustering, feedback structure) to performance.

**Weaknesses:**

- All experiments are conducted on powerful proprietary LLMs, I wonder whether the proposed method remains effective for opensource LLMs such as LLaMA or Qwen.
- It seems that the proposed method underperforms DSPy on MedQA, while surpassing it on other benchmarks, what might be the reason?
- How to determine the number of clusters when constructing the minibatches? How does this affect the method's effectiveness?

**Questions:**

See weaknesses.

---

> ### Author Response · Authors · 2024-11-22
>
> Thank you for reading the paper carefully, for the helpful feedback and the questions.
>
> > All experiments are conducted on powerful proprietary LLMs, I wonder whether the proposed method remains effective for opensource LLMs such as LLaMA or Qwen.
>
> Thanks for this question. We tried UniPrompt with Llama3 8B Instruct and Llama3 70B Instruct as the solver model on the ARC-challenge dataset, using one-line task description as the Init prompt compared to the baseline performance reported in the LLaMa model card. The results are summarized in the table below:
>
> | LLaMa Model | Baseline | UniPrompt |
> | ------------- | ------------- |  ------------- |
> | Llama3 8B Instruct | 82.4% | **91.7%** |
> | Llama3 70B Instruct | 94.4% | **95.8%** |
>
>
> > It seems that the proposed method underperforms DSPy on MedQA, while surpassing it on other benchmarks, what might be the reason?
>
> MedQA is a knowledge-heavy dataset from the medical domain. We suspect that may be a reason for few-shot DSPy prompt achieving the best performance. Among the zero-shot prompts, we observe that the DSPy prompt contained a summarized-reasoning instruction, "Consider the patient's symptoms, physical examination findings, and serologic results to identify" which was effective for the dataset.
>
> > How to determine the number of clusters when constructing the minibatches? How does this affect the method's effectiveness?
>
> The number of clusters is a hyperparameter. We had set it to 5 for our experiments. Based on your question, we also tried running UniPrompt with different number of clusters; the results are shown below.
>
> | Number of clusters | Accuracy |
> | ------------- | ------------- |
> | 2 | 85.81% |
> | 5 | 90.36% |
> | 10 | 87.82% |
>
> The appendix table (at the end of this response) containing the prompts shows that as the number of clusters increases, the feedback becomes more diverse and detailed (evident from the length of the prompt). However, this results in lower accuracy compared to having 5 clusters, which provides feedback that is both more concise and generalized, leading to better performance.

---

> > ### Author Response · Authors · 2024-11-22
> >
> > **Appendix**
> >
> > | Number of clusters | Prompt |
> > | ------------- | ------------- |
> > | 2 | _Given a query, you have to answer the MCQ question based on the given choices. Think step by step. If the answer is 0 option, output <Answer>0</Answer>. Ensure that your explanation accurately correlates with the correct scientific principles or logical reasoning relevant to the question and choices. Verify that the justification for the selected answer aligns with both the given context and the true answer, without making unsupported assumptions. Labelling Guidelines: 1. When answering questions that involve determining whether a behavior is learned or innate, ensure that your explanation directly correlates your chosen answer with specific evidence or context provided in the question. Provide clear reasoning that supports your conclusion, explicitly stating why the behavior is classified as learned or innate based on the details given. 2. Ensure that explanations not only state the correct answer but also provide a detailed rationale. This should include verifying how the student's choice aligns with the true answer, using logical reasoning and context. For instance, when explaining environmental impacts, explicitly describe how an action (like releasing heated water) affects the ecosystem, such as threatening aquatic organisms by altering temperature levels, rather than maintaining balance or increasing nutrients. This approach avoids assumptions and strengthens understanding_ |
> > | 5 | _Given a query, you have to answer the MCQ question based on the given choices. Think step by step. If the answer is 0 option, output <Answer>0</Answer>. Guidelines: 1) Clearly understand that oxygen is critical for combustion. Recognize that a candle in an open environment will have ample oxygen supply compared to one placed in a vacuum or covered by jars. This principle is crjars. This principle is crucial for determining outcomes related to combustion scenarios. 2) Ensure the explanation provided explicitly connects the chosen answer to the core reason behind the observed phenomenon. 3) Align the explanation logically with the direction of movement described in the question. 4) Accurately reflect the characteristics of each option provided in the choices. 5) Avoid assuming factors not directly correlated to the observed outcome. 6) After conducting experiments, especially those involving staining agents or chemical residues, it is important to consider the contamination and usability of materials. Used materials that are stained or have abhat are stained or have absorbed substances should generally be disposed of to prevent cross-contamination and ensure safety. 7) When identifying renewable energy resources that do not degrade the environment, focus on sources that are naturally replenished and have minimal environmental impact. In the context of Las Vegas, solar energy i Las Vegas, solar energy is a renewable and environmentally friendly option. 8) Ensure explanations clearly identify and address any misconceptions or incorrect assumptions made in the student's reasoning process. The explanation should contrast the chosen answer effectively with incorrect ffectively with incorrect choices, highlighting the unique factors that make the correct option stand out. 9) Ensure explanations accurately distinguish between physical and chemical changes, clarifying the defining characteristics of each to correctly identify evidence of a chemical reaction. 10) Ensure that the explanation explicitly addresses why the correct answer is chosen by emphasizing the key property or characteristic that directly answers the question. 11) Ensure that the explanation demonstrates comprehension of the biological processes involved, such as conjugation, and accurately distinguishes between the capabilities of different organisms in the context of tanisms in the context of the question_ |

---

> > > ### Author Response · Authors · 2024-11-22
> > >
> > > | Number of clusters | Prompt |
> > > | ------------- | ------------- |
> > > | 10 | _Given a query, you have to answer the MCQ question based on the given choices. Think step by step. If the answer is 0 option, output <Answer>0</Answer>. Ensure that the choice number matches the correct answer provided in the true answer field, as this is critical for accuracy. Each selected choice must be cross-verified with the true answer provided, and the reasoning in the explanation should align with the true answer field to improve accuracy. Carefully distinguish between instinctual behaviors and inherited traits to avoid misclassification.\n\nLabelling Guidelines:\n\n1. Ensure choices explicitly identify the primary function or characteristic associated with each option. Focus on the primary role when asked about the main function of a system.\n\n2. Require students to clearly differentiate between learned, instinctual, and inherited traits, especially in biology-related questions to avoid misclassification.\n\n3. For questions about anatomical motion, ensure the explanation explicitly assesses the primary action associated with each choice and compares it with the true anatomical motion. Focus on actions that primarily involve flexion or extension, such as bending or straightening, rather than lateral or unrelated motions. Strictly output the number of the correct choice.\n\n4. When identifying actions involving the elbow joint, determine if the primary motion is flexion or extension at the elbow. Focus on actions where the elbow joint is the central pivot point for the movement.\n\n5. Ensure that when identifying examples of unicellular organisms, the chosen example must demonstrate a single entity performing all necessary life functions independently, without the aid or cooperation of others. This example should not rely on the sequence or number of options provided but should clearly illustrate singularity in its actions, akin to how a unicellular organism functions entirely on its own. For instance, a correct example would be a single bacterium carrying out all life processes on its own, whereas multiple students working together would not represent a unicellular organism.\n\n6. Ensure that when selecting an answer, the choice number in your response matches exactly with the choice number provided in the question. Misalignment between the choice number in the question and your selected answer will lead to incorrect responses.\n\n7. **Rule 7: Ensure Logical Coherence and Accuracy in Explanations**\n\n   When answering questions, it is crucial to validate that your explanation is logically coherent with the fundamental principles involved and accurately corresponds to the correct choice number provided in the question. This includes:\n\n   - **Alignment with Core Concepts:** Make sure that your explanation aligns with the established scientific or mathematical principles. For example, in questions involving forces, refer to Newton’s Third Law of Motion which states that for every action, there is an equal and opposite reaction.\n\n   - **Accurate Choice Reference:** Double-check that the explanation you provide matches the choice number of the correct answer in the given options list. Misalignments can lead to incorrect answers even if the explanation is technically correct.\n\n8. Cross-verify your explanation with the true answer to avoid any misalignment, particularly focusing on the correctness of the choice number.\n\n9. When analyzing the direction of an object's movement under multiple forces, consider how each force vector contributes to the overall resultant direction. Specifically, identify the direction of each force and how they combine vectorially to produce the observed movement.\n\n10. When dealing with stained items, such as used paper cups from experiments, always assess the material composition and consult local recycling guidelines to determine if they can be recycled. Stained paper cups are often not recyclable due to contamination, so they should be disposed of in the trash unless local guidelines specify otherwise.\n\n11. Always consider the biological principles behind the options provided. For questions involving evolutionary changes, focus on factors like reproduction rate and genetic variance rather than physical attributes or unrelated characteristics._ |

---

> > > > ### Author Response · Authors · 2024-11-25
> > > > **Request for Feedback on the rebuttal**
> > > >
> > > > Dear Reviewer **SD4J**,
> > > >
> > > > Thank you again for your helpful feedback. We have done our best to address your concerns with additional experiments using Llama and different minibatch sizes.
> > > >
> > > > Please let us know if you have any further questions or comments after reading our rebuttal.

---

> > > > ### Comment · Reviewer_SD4J · 2024-11-26
> > > > **Response to Authors**
> > > >
> > > > I appreciate the author's efforts in their rebuttal. I still have the following concerns:
> > > >
> > > > - I still think that more models are needed to support the effectiveness of the proposed method. I suggest the authors incorporate more LLMs in their next version.
> > > >
> > > > - Regarding the experiments using different number of clusters, what is the benchmark used? And what is the result reported in the main paper? How many clusters are used?

---

> ### Author Response · Authors · 2024-11-27
>
> > I still think that more models are needed to support the effectiveness of the proposed method. I suggest the authors incorporate more LLMs in their next version.
>
> Thank you for your feedback. We believe we have conducted extensive experiments, given the computational resources available. Many of the baselines we compare against, such as Protegi, Evoke, TextGrad, and OPRO, also focus on evaluating capable frontier models at that time, including GPT-3.5, GPT-4, and PaLM2. Following a similar approach, we have included these models in our paper and, during this rebuttal, have also provided an ablation study with LLama3.
>
> > Regarding the experiments using different number of clusters, what is the benchmark used? And what is the result reported in the main paper? How many clusters are used?
>
> The benchmark for the experiments is the ARC-Challenge dataset. The main paper reports results using 5 clusters, achieving an accuracy of 86.0% with a beam size of 2 and 90.5% with greedy decoding.The results shared earlier in the rebuttal correspond to greedy decoding, aligning closely with the paper’s reported value of 90.5% (previous table: 90.36%), with only minor differences.

---

### Official Review · Reviewer_Gcge · 2024-11-01

**Soundness:** 3
**Presentation:** 3
**Contribution:** 3
**Rating:** 5
**Confidence:** 3

**Summary:**

This paper proposes a prompt optimization method $U_{NI}P_{ROMPT}$ to generate complex prompts for covering multiple facets of a task and improving overall accuracy. Its effectiveness is demonstrated through relevant experiments.

**Strengths:**

The strategy for creating mini-batches to generate edits and then aggregating them at the batch level to yield the final edit in a feedback mechanism is interesting.

Experimental results achieves SOTA.

**Weaknesses:**

1.The paper contains some detailed issues, e.g., $U_{ni}P_{rompt}$ mentioned in  line 100-101; an extra space before "So" mentioned in line 197-198.

2. The experiments should be re-organized and polished carefully before submission, especially in Section 4.1,  4.2 and 4.5.

3. Some qualitative experiments are supported be added to validate the differences in report accuracies under various mini-batch sizes.

**Questions:**

The impact of mini-batch size.

The impact of aggregation.

---

> ### Author Response · Authors · 2024-11-22
>
> Thank you for reading the paper carefully, for the helpful feedback and the questions.
>
> > 1. The paper contains some detailed issues, e.g. UniPrompt, mentioned in line 100-101; an extra space before "So" mentioned in line 197-198.
>
> Thank you for pointing this out. We will fix them in the updated version.
>
> > 2. The experiments should be re-organized and polished carefully before submission, especially in Section 4.1, 4.2 and 4.5.
>
> Thanks for the feedback. We will update these sections for better clarity.
>
> > The impact of mini-batch size.
>
> To assess the impact of mini-batch size, we ran UniPrompt with different mini-batch sizes on the ARC dataset. Here are the results:
>
> | Mini batch size | Accuracy |
> | ------------- |  ------------- |
> | 2 | 84.90% |
> | 5 | 90.36% |
> | 8| 91.15% |
>
> (Table 1 ARC corresponds to mini batch of size 3. For the experiments in the paper, we didn't do extensive cross-validation for this parameter, and as we mention in Line 402, we tuned over mini-batch sizes {3,4,5} for the datasets.)
>
> With increase in mini-batch size, we observe an increase in accuracy. That said, it is a hyperparameter hence there will be an optimal number for each dataset. The mini-batch size affects the feedback based on the wrong examples that are obtained in each round.
>
> > 3. Some qualitative experiments are supported be added to validate the differences in report accuracies under various mini-batch sizes.
>
> As we can observe in the appendix table (at the end of this response) table, with mini-batch size as 2 there are four points in the guidelines mentioned which are specific to a few examples from the dataset. With mini-batch size as 5, the prompt is more detailed with 11 instructions and guidelines. At last, with mini batch size as 8, the prompt contains fewer guidelines compared to the previous prompts but they are more generalizable and hence attain higher accuracy.
>
> > The impact of aggregation.
>
> To evaluate the effect of aggregation, we conducted an ablation study. When tested on the ARC dataset, using aggregation resulted in an accuracy of 90.36%, whereas omitting aggregation led to a reduced accuracy of 84.07%. Aggregation proves beneficial by preserving the most valuable feedback and enabling the generation of broader task facets in the refined prompt.
>
> **Appendix**
> | Mini batch size | Prompt |
> | ------------- |  ------------- |
> | 2 |  _Given a query, you have to answer the MCQ question based on the given choices. Think step by step. If the answer is 0 option, output <Answer>0</Answer>. Ensure that the explanation and chosen answer align with the reasoning provided; verie reasoning provided; verify that the explanation directly supports the selection of the answer, and if the explanation identifies a choice as correct, it should be the one selected. Labelling Guidelines: 1) Ensure that the explanation provided accurately reflects scientific principles or factual information related to the question. 2) Ensure that the explanation accurately reflects the ecological relationships and roles of each animal in the ecosystem, identifying prey-predator dynamics correctly. 3) Ensure that the explanation accurately differentiates between the reproductive methods of the organisms mentioned in the question, with a particular focus on distinguishing between asexual and sexual reproductioual and sexual reproduction, and clearly stating which organisms can undergo conjugation, such as recognizing that a paramecium can conjugate, but an amoeba cannot. 4) Ensure that explanations accurately distinguish between elements and compounds, specifically recognizing that if a substance cannot be broken down into simpler substances, it is an element, not a compound. This ent, not a compound. This distinction is crucial for correctly identifying substances in questions related to chemical composition_

---

> > ### Author Response · Authors · 2024-11-22
> >
> > **Appendix(continued)**
> > | Mini batch size | Prompt |
> > | ------------- | ------------- |
> > | 5 | _Given a query, you have to answer the MCQ question based on the given choices. Think step by step. If the answer is 0 option, output <Answer>0</Answer>. Guidelines: 1) Clearly understand that oxygen is critical for combustion. Recognize that a candle in an open environment will have ample oxygen supply compared to one placed in a vacuum or covered by jars. This principle is crjars. This principle is crucial for determining outcomes related to combustion scenarios. 2) Ensure the explanation provided explicitly connects the chosen answer to the core reason behind the observed phenomenon. 3) Align the explanation logically with the direction of movement described in the question. 4) Accurately reflect the characteristics of each option provided in the choices. 5) Avoid assuming factors not directly correlated to the observed outcome. 6) After conducting experiments, especially those involving staining agents or chemical residues, it is important to consider the contamination and usability of materials. Used materials that are stained or have abhat are stained or have absorbed substances should generally be disposed of to prevent cross-contamination and ensure safety. 7) When identifying renewable energy resources that do not degrade the environment, focus on sources that are naturally replenished and have minimal environmental impact. In the context of Las Vegas, solar energy i Las Vegas, solar energy is a renewable and environmentally friendly option. 8) Ensure explanations clearly identify and address any misconceptions or incorrect assumptions made in the student's reasoning process. The explanation should contrast the chosen answer effectively with incorrect ffectively with incorrect choices, highlighting the unique factors that make the correct option stand out. 9) Ensure explanations accurately distinguish between physical and chemical changes, clarifying the defining characteristics of each to correctly identify evidence of a chemical reaction. 10) Ensure that the explanation explicitly addresses why the correct answer is chosen by emphasizing the key property or characteristic that directly answers the question. 11) Ensure that the explanation demonstrates comprehension of the biological processes involved, such as conjugation, and accurately distinguishes between the capabilities of different organisms in the context of tanisms in the context of the question_
> > | 8 | _Given a query, you have to answer the MCQ question based on the given choices. Think step by step. If the answer is 0 option, output <Answer>0</Answer>. Guidelines: 1) When identifying chemical similarities, always verify the group number of each element from the periodic table. Do not assume based on partial information or incorrect assumptions. 2) Ensure that explanations and answers reflect accurate scientific definitions and principles.- Understand that carbohydrates, like fruit, provide the most rapid energy release compared to other food types. - Recognize that Earth rotates once on its axis each day. - Acknowledge that a mixture consists of components that retain their original properties and can be physically separated, unlike chemical compounds. - Apply Newton's third law: the force exerted by the tree on the student is equal and opposite to the force exerted by the student on the tree. - Identify that both astronomers and biologists use optical devices such as telescopes and microscopes to make discoveries. 3) Ensure that answers and explanations accurately reflect the true function or characteristic being tested, and review the question carefully to match key concepts with the correct choice. 4) Ensure explanations provide a clear rationale that logically supports the chosen answer, explicitly addressing why the other options are less suitable. Consider how environmental factors impact outcomes, such as the presence or absence of air in scenarios involving physical conditions. 5) Ensure that explanations not only identify the correct choice but also detail the distinct functions or characteristics that differentiate the correct choice from the others, by using specific examples and definitions. This helps in understanding how each choice relates to the question's core concept._

---

> > > ### Author Response · Authors · 2024-11-25
> > > **Request for Feedback on the rebuttal**
> > >
> > > Dear Reviewer **Gcge**,
> > >
> > > Thank you again for your helpful feedback. We’ve done our best to address your concerns with additional experiments and qualitative results.
> > >
> > > Please let us know if you have any further questions or comments after reading our rebuttal.

---

> > > ### Comment · Reviewer_Gcge · 2024-12-03
> > >
> > > Thanks for the response. I will keep my score at this moment.

---

> > > > ### Author Response · Authors · 2024-12-03
> > > > **Thanks**
> > > >
> > > > Dear reviewer,
> > > > Given that the discussion window is closing soon, it would be very helpful if you could let us know why you are still on the fence about the paper. It would help us prepare the next version better, in the worst case. Thanks again, for your time and feedback!

---

### Author Response · Authors · 2024-12-04
**Summary of rebuttal discussion**

Dear reviewers and AC,

We summarize the reviews, the questions during the discussion period, and our rebuttal below.

**Strengths pointed out by the reviewers**: All the reviewers point out the **two-tier feedback component** (mini-batching and batching) of UniPrompt as a strength. All the reviewers also appreciated the reasonably comprehensive empirical evaluation of the work, ablations validating the contributions, and asked for some additional evaluations/ablations during the rebuttal. All the reviewers also recognize that UniPrompt **outperforms the state-of-the-art methods**. In general, everyone has given a positive score (3) on the technical contributions and soundness of the ideas.

**Concerns raised**: The primary concerns were regarding ablations (1-2) and generalization aspects (3-4): (1) mini batching, (2) aggregation/clustering, (3) use of open source LLMs like LLaMA, (4) experiments on BBH benchmark which is popular in the literature.

**Our rebuttal responses/results**:

(1) With increase in mini-batch size, we observe an increase in accuracy (as we show on the ARC dataset). It is a hyperparameter that can be tuned for each dataset.

(2) When tested on the ARC dataset, using aggregation resulted in an accuracy of **90.36%**, whereas omitting aggregation led to a reduced accuracy of 84.07%. Aggregation proves beneficial by preserving the most valuable feedback and enabling the generation of broader task facets in the refined prompt.

(3) On the ARC-challenge dataset, UniPrompt with Llama3 8B Instruct as the solver model yields **91.67%** accuracy (using one-line task description as the Init prompt) compared to the baseline performance reported in their model card which is 82.4%. Similarly, for Llama3 70B Instruct as the solver model, UniPrompt achieves **95.58%** accuracy as against the baseline accuracy of 94.4% reported on the model card. These results, along with the results in the paper, show that UniPrompt generalizes well to open-source as well as closed-source models, and across model sizes.

(4) From the [table](https://openreview.net/forum?id=ViRDmDAfjg&noteId=E3NQWM3fjM) we provide in our rebuttal regarding the BBH benchmark, it is evident that UniPrompt shows a significant improvement over OPRO for all tasks except one. For instance, it achieves significantly higher accuracy in tasks such as "_Boolean Expression_" (**92.37%** vs. 78.74%), "_Date Understanding_" (**81.96%** vs. 52.59%), and "_Navigate_" (**77.16%** vs. 51.74%). UniPrompt's prompts are also much more informative compared to that of OPRO and other baselines.

These results, along with the results in the paper, show that UniPrompt generalizes well across task scenarios (fact checking, reasoning,), domains (math, code), and types (generative, discriminative).

---

### Meta-Review · Area_Chair_8ToW · 2024-12-26

**Metareview:**

This paper proposed a new prompt optimization algorithm, UniPrompt, which exploits the structure in the prompt optimization, uses clustering to capture different facet of the task feedback and feedback mechanism to capture generalizable facets. The experiment shows the effectiveness of the proposed method comparing with other prompt optimization algorithms. Reviewers all appreciate the contribution of the work; however, the concerns still persists on the ablation studies such as different base LLMs, prompt initialization, and clustering. Addressing these concerns through additional ablation studies would strengthen the paper and provide a more comprehensive understanding of UniPrompt's capabilities and limitations.

**Additional Comments On Reviewer Discussion:**

Results on new datasets are added during rebuttal, along with some ablation study such as different mini-batch size. The concerns still persists on the ablation studies such as different base LLMs, prompt initialization, and clustering.

---

### Decision · Program_Chairs · 2025-01-22

Reject